# Scaling of sensory information in large neural populations shows signatures of information-limiting correlations

MohammadMehdi Kafashan[1], Anna W. Jaffe [1], Selmaan N. Chettih[1], Ramon Nogueira[2], Iñigo Arandia-Romero [3,4], Christopher D. Harvey[1], Rubén Moreno-Bote[5,6] & Jan Drugowitsch [1 ✉]

How is information distributed across large neuronal populations within a given brain area? Information may be distributed roughly evenly across neuronal populations, so that total information scales linearly with the number of recorded neurons. Alternatively, the neural code might be highly redundant, meaning that total information saturates. Here we investigate how sensory information about the direction of a moving visual stimulus is distributed across hundreds of simultaneously recorded neurons in mouse primary visual cortex. We show that information scales sublinearly due to correlated noise in these populations. We compartmentalized noise correlations into information-limiting and nonlimiting components, then extrapolate to predict how information grows with even larger neural populations. We predict that tens of thousands of neurons encode 95% of the information about visual stimulus direction, much less than the number of neurons in primary visual cortex. These findings suggest that the brain uses a widely distributed, but nonetheless redundant code that supports recovering most sensory information from smaller subpopulations.

[1] Department of Neurobiology, Harvard Medical School, Boston, MA 02115, USA. [2] Center for Theoretical Neuroscience, Mortimer B. Zuckerman Mind Brain Behavior Institute, Columbia University, New York, NY, USA. [3] ISAAC Lab, Aragón Institute of Engineering Research, University of Zaragoza, Zaragoza, Spain. [4] IAS-Research Center for Life, Mind, and Society, Department of Logic and Philosophy of Science, University of the Basque Country, UPV-EHU, Donostia-San Sebastián, Spain. [5] Center for Brain and Cognition and Department of Information and Communication Technologies, Universitat Pompeu Fabra, Barcelona, Spain. [6] Serra Húnter Fellow Programme and ICREA Academia, Universitat Pompeu Fabra, Barcelona, Spain. ✉email: jan_drugowitsch@hms.harvard.edu

Our brains encode information about sensory features in the activity of large neural populations. The amount of encoded information provides an upper bound on behavioral performance, and so exposes the efficiency and structure of the computations implemented by the brain. The format of this encoding reveals how downstream brain areas ought to access the encoded information for further processing. For example, the amount of information in visual cortex about the drift direction of a moving visual stimulus determines how well one could in principle discriminate different drift directions if the brain operates at maximum efficiency, and its format tells us how downstream motion-processing areas ought to "read out" this information. Therefore, knowing how the brain encodes sensory information about the world is necessary if we are to understand the computations it performs. Unfortunately, we still know little about how sensory information is distributed across neuronal populations even within a single brain area. Is information spread evenly and largely independently across neurons, or in a way that introduces significant redundancy? In the first scenario, one would need to record from the whole neuronal population to get access to all available information, whereas in the second scenario only a fraction of neurons would be needed.

The amount of information about a stimulus feature that can be extracted from neural population activity depends on how this activity changes with a change in the stimulus feature. For information that can be extracted by a linear decoder, which is the information we focus on in this work, it depends on the neurons' tuning curves, as well as how their activity varies across repetitions of the same stimulus (i.e., "noise")[1–4]. Due to the variability in neural responses to repetitions of the same stimulus, each neuron's response provides limited information about the stimulus feature[5–9]. If the noise is independent across neurons, it can be averaged out by pooling across neurons[10], and total information would on average increase by the same amount with every neuron added to this pool (Fig. 1a, red). This corresponds to the first scenario in which information is spread evenly across neurons. If, however, the trial-to-trial variations in spiking are shared across neurons—what are referred to as "noise correlations"—the situation is different. In general, depending on their structure, noise correlations can either improve or limit the

amount of information (Fig. 1b), such that the presence of correlated noise alone does not predict its impact. In a theoretical population with translation-invariant tuning curves (i.e., the individual neurons' tuning curves are shifted copies of each other) and noise correlations that are larger for neurons with similar tuning, information might quickly saturate with population size[10,11], corresponding to the second scenario (Fig. 1a, black). Even though such correlation structures, which are traditionally studied in sensory areas, have been observed across multiple brain areas[10,12–15], neural tuning is commonly more heterogeneous than assumed by Zohary et al.[10]. A consequence of this heterogeneity is that sensory information might grow without bound even with noise correlations of the aforementioned structure[16]. Overall, it remains an open question if sensory information saturates in large neural populations of human and animal brains[1].

If information saturates in such populations, then, by the theory of information-limiting correlations (TILC)[17], information in large populations is limited exclusively by one specific component of the noise correlations. This component introduces noise in the direction of the change of the mean population activity with stimulus value (e.g., drift direction; black arrow in Fig. 1b, bottom), thus limiting information about this value. Measuring this noise correlation component directly in neural population recordings is difficult, as noise correlations are, in general, difficult to estimate well[18], and the information-limiting component is usually swamped by other types of correlations that do not limit information[17,19]. Fortunately, however, TILC also predicts *how* information scales with population size if information-limiting correlations are present. We thus exploited this theory to detect the presence of information-limited correlations indirectly by examining how information scales with population size.

In this work, we search for the presence of information-limiting correlations, by simultaneously recording the activity of hundreds of neurons in V1 of awake mice in response to drifting gratings, with hundreds of repeats of each stimulus. We asked how these neurons encoded information about the direction of the moving visual stimulus. We found that noise correlations reduce information even within the limited neural populations we

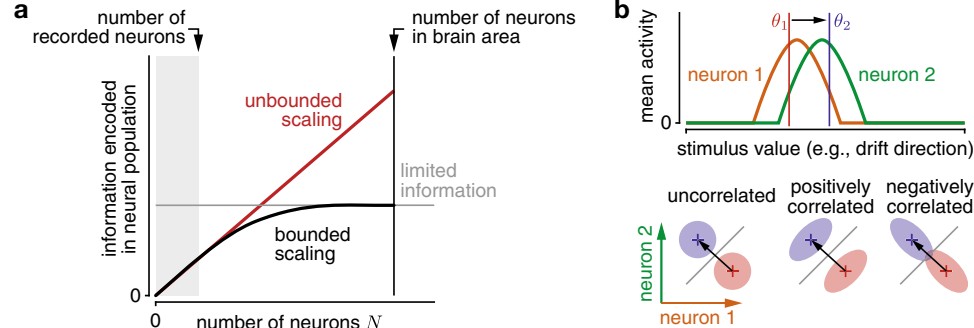

**Fig. 1 Information scaling in large neural populations, and the impact of noise correlations on information. a** The information that a population of neurons can encode about some stimulus value is always a non-decreasing function of the population size. Information might on average increase with every added neuron (unbounded scaling; red) if the information is evenly distributed across all neurons. In contrast, information can rapidly saturate if information is redundant, and thus it is not strictly limited by population size, but by other factors. In general, it has only been possible to record from a very small subset of neurons of a particular area (gray shaded), from which it is hard to tell the difference between the two scenarios if the sampled population size is too small. **b** The encoded information is modulated by noise correlations. This is illustrated using two neurons with different tunings to the stimulus value (top). The amount of information to discriminate between two stimulus values ($\theta_1$/red and $\theta_2$/blue) depends on the difference in mean population activity (crosses) between stimuli, and the noise correlations (shaded ellipsoids) for either stimulus (bottom, showing joint neural activity of both neurons). The information is largest when the noise is smallest in the direction of the mean population activity difference (black arrow), which leads to the largest separation across the optimal discrimination boundary (gray line). In this example, positive correlations boost information (middle), whereas negative correlations lower it (right), when compared to uncorrelated neurons (left). In general, the impact of noise correlations depends on how they interact with the population's tuning curves.

could record. Applying TILC to compartmentalize information-limiting correlations from nonlimiting correlations, and to extrapolate the growth of information to larger neural populations, we found that on the order of tens of thousands of neurons would be required to encode 95% of the information about the direction of the moving stimulus. Given that there are hundreds of thousands of neurons in this brain region, this means that only a small fraction of the total population is needed to encode this information. This is not because only a small fraction of neurons contains information about the stimulus; rather, we found that most neurons contain information about the stimulus, but because information is represented redundantly, only a small fraction of these neurons is actually needed. Notably, the size of the required neural population depends only weakly on stimulus contrast; thus, increasing the amount of information in this brain area does not substantially increase the number of neurons required to encode 95% of the information about the stimulus. Finally, we found that the low-dimensional neural subspace that captures a large fraction of the noise correlations does not encode a comparably large fraction of information. Overall, our results suggest that information in mouse V1 is both highly distributed and highly redundant, which is true regardless of the total amount of information encoded.

## Results

**Neural response to drift direction of moving visual stimuli.** To measure how sensory information scales with population size, we used two-photon calcium imaging to record neural population activity from layer 2/3 of V1 in awake mice observing a low-contrast drifting grating (10% contrast). The drift direction varied across trials, with each trial drawn pseudorandomly from eight possible directions, spaced evenly around the circle (Fig. 2a). We simultaneously recorded 273–386 neurons (329 on average) across four mice and a total of 16 sessions (Fig. 2b), and analyzed temporally deconvolved calcium activity, summed up over the stimulus presentation period as a proxy for their spike counts within that period. The tuning curves of individual neurons (Fig. 2c) revealed that, on average, only a small fraction of neurons (5–45% across mice/sessions, 18% average) were tuned to the grating's drift direction, while a larger fraction of neurons (38–60% across mice/sessions, 48% average) were sensitive to the grating's orientation, but not its direction of drift. The remaining neurons had no appreciable tuning (14–52% across mice/sessions, 34% average), but were nonetheless included in the analysis, as they can contribute to the information that the population encodes through noise correlations[20,21]. See Supplementary Figs.1–3 for more examples of neural responses, tuning curves, pairwise noise correlations, and raw calcium traces. We found no

significant impact of the drift direction in the previous trial on neural responses in the current trial (Supplementary Fig. 1b and Supplementary Table 1). Tuning curves were plotted for the sole purpose of characterizing individual neural responses, but our fits had no bearing on any of our further analysis.

**Noise correlations limit information.** To quantify stimulus information encoded in the response of neural populations, we asked how well a linear decoder of the recorded population activity (i.e., information decodable by a single neural network layer) would allow us to discriminate between a pair of drift directions (Fig. 3a). Importantly, our aim was to measure information that population activity conveyed about drift direction in general, without prioritizing specific drift directions over others. Even though subselecting a limited set of drift directions is common in animal training, we here focused on discriminating drift directions in pairs only as a tool to get at information about drift direction in general, which should be more reflective of real-world demands. We measured the decoder's performance by generalizing linear Fisher information, usually restricted to fine discriminations, to coarse discrimination (Fig. 3b). This generalization is closely related to the sensitivity index $d'$ from signal detection theory[3,22], and has a set of appealing properties (see "Methods"). In particular, combining the activity of two uncorrelated neural populations causes their associated Fisher information to add, so that it does not trivially saturate like other measures of discrimination performance (Fig. 3c, inset).

We used generalized Fisher information to measure how information about drift direction scales with the number of neurons in the recorded population. Because this scaling depends on the order in which we add particular neurons to the population (individual neurons might contribute different amounts of additional information to a population), we measured average scaling by averaging across a large number of different random orderings (see "Methods"). Figure 3c shows this average scaling for one example session for discriminating between drift directions of 135° and 180° (arbitrary choice; as shown below, other drift direction combinations resulted in comparable information scaling). Information increases with population size, but, on average, additional neurons contribute less additional information to larger populations than to smaller ones. The resulting sublinear scaling is expected if noise correlations limit information. Indeed, trial-shuffling the data to remove pairwise correlations resulted in information that scaled linearly, with average information exceeding that of the non-shuffled data for all population sizes except, trivially, for single neurons, and a significantly higher total information within the recorded population (bootstrap, $p \approx 0.0062$). Such linear scaling was not

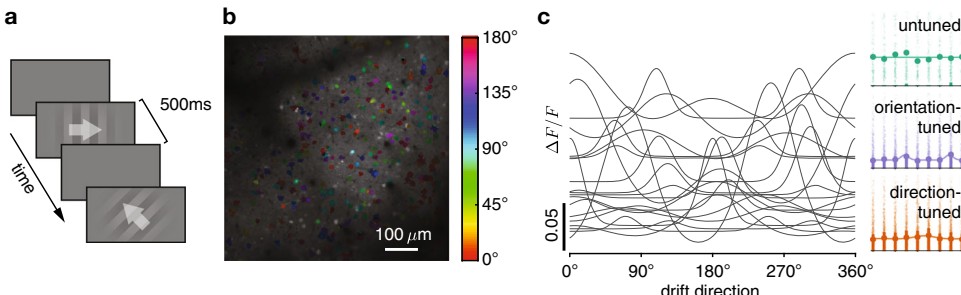

**Fig. 2 Experimental design, population recordings, and neural tuning. a** Mice passively observed sequences of drifting gratings (white arrows overlaid for illustration only), interleaved with blank screens. **b** Example field-of-view with significantly tuned neurons color coded by their preferred orientation tuning. **c** Left: example fitted tuning curves of 20 significantly tuned neurons. Right: example tuning curves (dots + bars: raw tuning, mean ± 25–75% percentiles; line: fitted) fitted to per-trial neural responses (dots, horizontally jittered) for an untuned (top), orientation-tuned (middle) and direction-tuned (bottom) neuron.

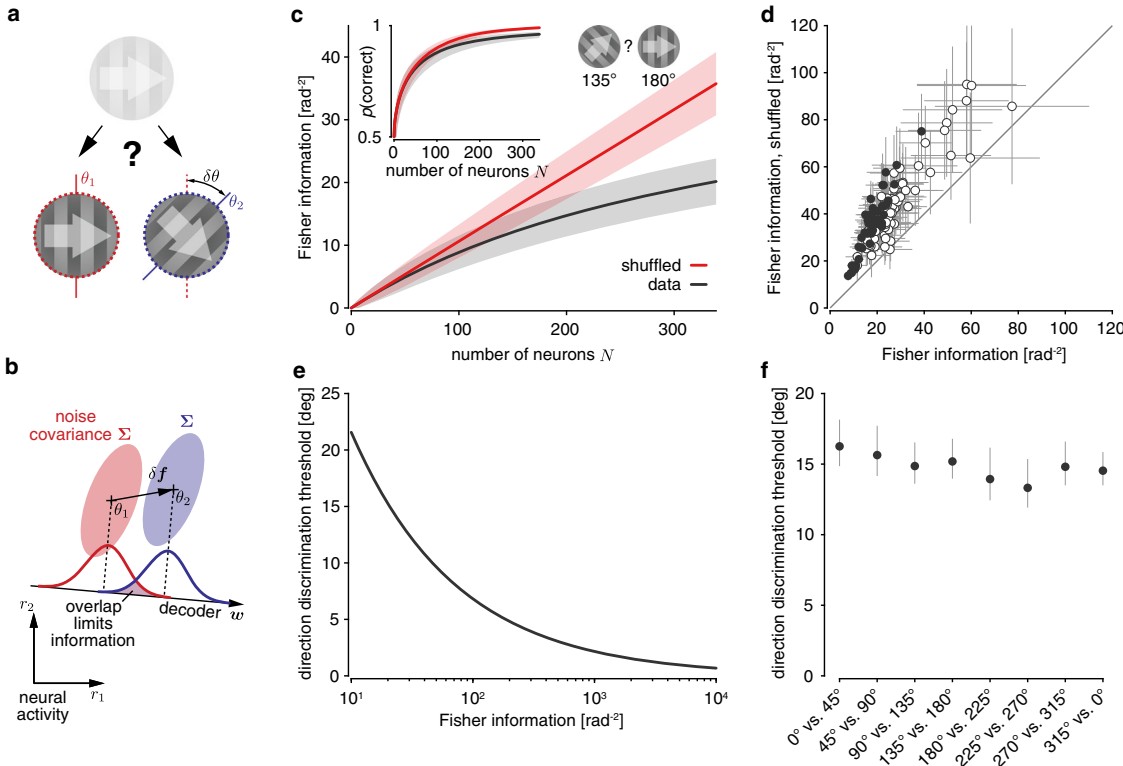

**Fig. 3 Noise correlations limit information across all drift directions. a** A drift direction discrimination task, in which a hypothetical observer needs to judge which of two template drift directions ($\theta_1$ or $\theta_2$; indicated by white arrows) an observed low-contrast drifting grating corresponds to. **b** Mean activity $f$ (crosses) and noise covariance $\Sigma$ (shaded area $\approx$ 2SDs) of a pair of neurons across repeated presentation of the same two drift directions, $\theta_1$ (red) and $\theta_2$ (blue). Linear information about drift direction is limited by the projection of the noise onto the optimal linear decoder $w$. This decoder depends on how mean activity changes with drift direction ($\delta f = f(\theta_2) - f(\theta_1)$) and the noise covariances $\Sigma$. **c** The information associated with discriminating between drift directions 135° and 180° scales sublinearly with population size (black; mean ± 1 SD across random orderings of neurons within the population). If we remove noise correlations by shuffling trials across neurons, the information scales linearly (red). This linear growth would not be apparent from the probability of correctly identifying the stimulus' drift direction (inset), which is monotonically, but non-linearly related to Fisher information, and saturates in both cases. **d** Information in the recorded population was consistently larger for trial-shuffled data across different discriminations, sessions, and mice. Each dot (mean ± 1 SD of information estimate; filled = significant increase, bootstrap, $p < 0.05$) shows the information estimated for one discrimination with $\delta\theta = 45°$. **e** The drift direction discrimination threshold (corresponding to 80% correct discriminations) we would expect to see in a virtual discrimination experiment drops with the amount of information that V1 encodes about drift directions. **f** The inferred drift direction discrimination threshold for the same session as in panel **c** is comparable across the different drift direction pairs with $\delta\theta = 45°$ used to estimate Fisher information with the recorded population.

apparent if we measured discrimination performance by the fraction of correct discriminations (Fig. 3c, inset), illustrating the point that Fisher information is indeed a better measure to analyze information scaling. Removing noise correlations resulted in a significant information increase in all our datasets (Fig. 3d; paired $t_{63} = -17.93$, two-sided $p \approx 1.96 \times 10^{-26}$; statistics computed across all sessions and mice, but only across non-overlapping $\delta\theta = 45°$ discriminations to avoid duplicate use of individual drift direction trials; see Supplementary Table 2 for avg. per-neuron information for all sessions/mice), confirming that noise correlations indeed limit information in our recorded populations.

To aid interpretation of the estimated amounts of Fisher information, we translated them into quantities that are more frequently measured in experiments. Specifically, we assumed that the recorded neural population was used to discriminate between two close-by drift directions in a virtual fine discrimination task (similar to Fig. 3a). For a given estimate of Fisher information, we could then determine the expected discrimination threshold at which the ideal observer could correctly discriminate between two drift directions in 80% of the trials based solely on neuronal responses (Fig. 3e). This resulted in a discrimination threshold of

~15.2° for the Fisher information estimated from a 135° vs. 180° discrimination (Fig. 3f). Previously reported discrimination threshold of mice, as measured from behavioral performance, ranged from 6.6°[23] over 10–20°[24], to 30–40°[25]. These numbers provide an orders-of-magnitude comparison, but cannot be directly compared to our estimate, as neither study exactly matched the stimuli we used. Moreover, previous work has shown that attending to a stimulus boosts the information encoded about this stimulus[26,27]. As our animals were passive observers that were not actively engaged in any task, the estimated threshold likely underestimate discrimination capabilities. Indeed, higher running speeds, which were previously used as a proxy for increased attention[28], resulted in increased information (as shown previously by Dadarlat and Stryker[29]) and lower thresholds (Supplementary Fig. 4). In line with previous findings[29], this information boost was caused by a combination of a change in population tuning, per-neuron noise variability, and pairwise noise correlations, rather than either of these factors in isolation (Supplementary Fig. 5). Overall, the estimated thresholds provide a reasonable interpretation of the information encoded in the recorded population. Computing the discrimination threshold for all drift direction pairs with $\delta\theta = 45°$ resulted in comparable

thresholds that did not differ significantly (bootstrap, two-sided $p \approx 0.50$ for session shown in Fig. 3f, two-sided $p > 0.49$ for all sessions/mice). We found comparable information across all drift directions, confirming that we recorded from populations that were homogeneously tuned across all drift directions.

**Neural signatures of limited asymptotic information.** To identify neural signatures of limited encoded information, we relied on the TILC that showed that noise correlations in large populations can be compartmentalized into information-limiting and nonlimiting components[17]. The limiting component is scaled by the inverse of the asymptotic information $I_\infty$, which is where information asymptotes in the limit of a large number of neurons[17,19]. This compartmentalization allowed us to split the information $I_N$ in a population of $N$ neurons into the contribution of limiting and non-limiting components (see "Methods"), resulting in

$$I_N = \frac{1}{\frac{1}{cN} + \frac{1}{I_\infty}}. \tag{1}$$

This expression assumes that the non-limiting component contributes $c$ information per neuron on average, irrespective of the current population size. Model comparison to alternative non-limiting component scaling models confirmed that this assumption best fits our data (Supplementary Fig. 6b).

Increasing the population size $N$ in Eq. (1) reveals how information ought to scale in small populations if it is limited in large populations (Fig. 1). Information would initially grow linearly, closely following $cN$. However, for sufficiently large $N$, it would start to level off and slowly approach the asymptotic information $I_\infty$. If we were to record from a small number of neurons, we might only observe the initial linear growth and would wrongly conclude that no information limit exists (Fig. 1). Therefore, simultaneously recording from sufficiently large populations is important to identify limited asymptotic information.

To distinguish between a population in which information does not saturate from one in which it does, we fitted two models to the measured information scaling. The first assumed that, within the recorded population, information scales linearly and without bound. We might observe this information scaling if, on average, each neuron contributes the same amount of information. The second model corresponds to Eq. (1), and assumes that information asymptotes at $I_\infty$. Our fits relied on a large number of repetitions (at least as many as the number of recorded neurons) of the same drift direction within each experimental session to ensure reliable, bias-corrected information estimates[30]. These estimates are correlated across different population sizes, as estimates for larger populations share data with estimates for smaller populations. Unlike previous work that estimated how information scales with population size[31–33], we accounted for these correlations by fitting how information increases with each additional neuron, rather than fitting the total information for each population size. This information increase turns out to be statistically independent across population sizes (see "Methods"), making the fits statistically sound and side-stepping the problem of fitting correlated data.

Figure 4a illustrates the fit of the limited-information model to the data of a single session. We fitted the average information increase with each added neuron (Fig. 4a, top), and from this predicted the total information for each population size (Fig. 4a, bottom). Bayesian model comparison to a model that assumed unbounded information scaling confirmed that a model with limited asymptotic information was better able to explain the measured information scaling (Watanabe–Akaike Information Criterion $WAIC_{unlim} = -529.25$ vs. $WAIC_{lim} = -531.59$; smaller is better). This was the case for almost all discriminations with $\delta\theta = 45°$ across sessions and mice (Supplementary Fig. 6a). Furthermore, the same procedure applied to the shuffled data resulted in better model fits for the unbounded information model, confirming that our model comparison was not a priori biased towards the limited-information model (Supplementary Fig. 6a). Two sets of simulations with idealized and realistic neural models further confirmed that this model comparison was able to recover the correct underlying information scaling (Supplementary Fig. 7). Therefore, information about drift direction is limited in the neural population responses within our dataset.

This result of limited drift direction information was corroborated by a second analysis. We start by observing that Eq. (1) can be rewritten as $1/I_N = a(1/N) + 1/I_\infty$, which is linear in the inverse population size $1/N$ with slope $a = 1/c$. Increasing the population size, $N \to \infty$, causes the inverse information to approach the asymptotic information, $1/I_N \to 1/I_\infty$. Therefore, we can distinguish between limited asymptotic information and unbounded information scaling (i.e., $I_\infty \to \infty$) by plotting $1/I_N$ against $1/N$, and estimating its intercept at $1/N \to 0$. A non-zero intercept confirms limited asymptotic information, whereas a zero intercept would suggest information to scale without apparent bounds. When we analyzed the previous single-session data, we found that the inverse information indeed tightly scales linearly with the information population size (linear regression, adjusted $R^2 \approx 1$), as predicted by the model (Fig. 4b). Furthermore, the intercept at $1/N \to 0$ was significantly above zero (linear regression, $\beta_0 \approx 0.023$, two-sided $p < 10^{-6}$), suggesting that information saturates with $N$. We found comparably good linear fits for all sessions/mice across all $\delta\theta = 45°$ discriminations (average adjusted $R^2 \approx 0.999$; Supplementary Fig. 8a), and intercepts that were all significantly above zero ($\beta_0 \approx 0.023$, $t_{63} = 17.95$, two-sided $p < 10^{-10}$ across non-overlapping discriminations; Supplementary Fig. 8b), confirming the results of our model comparison.

In addition to supporting the distinction between information-limited and unbounded information scaling, TILC also allowed us to estimate the magnitude at which information would asymptote if we increased the population size beyond that of our recorded population. This is a theoretical measure that would be reached only for infinitely large virtual populations that have the same statistical structure as the recorded neurons. Despite this limitation, it gives insight into the order of magnitude of the information that we could expect to be encoded in the large populations of neurons present in mammalian cortices. To quantify the uncertainty associated with extrapolations beyond observed population sizes, we relied on Bayesian model fits that provide posterior distributions over our estimates of $I_\infty$, as illustrated in Fig. 4c. These posteriors were comparable across the discrimination of different drift direction pairs (Fig. 4d). Comparable information estimates across different drift direction pairs were essential to make these estimates meaningful, as different estimates would have implied that these estimates are driven by neural subsets within a heterogeneous population rather than being a statistical property of the whole population, as desired. Furthermore, it allowed us to reduce our uncertainty in the $I_\infty$ estimates by pooling the fits across different, non-overlapping drift direction pairs (Fig. 4d; gray). Indeed, Bayesian model comparison that accounts for the larger number of parameters of multiple individual per-discrimination fits confirmed that those were outperformed by pooled fits for all but two experimental sessions across all tested drift direction differences (Supplementary Fig. 9). This provided further evidence that, for a fixed drift direction difference, the measured information scaling was statistically indistinguishable across different discriminations within each session.

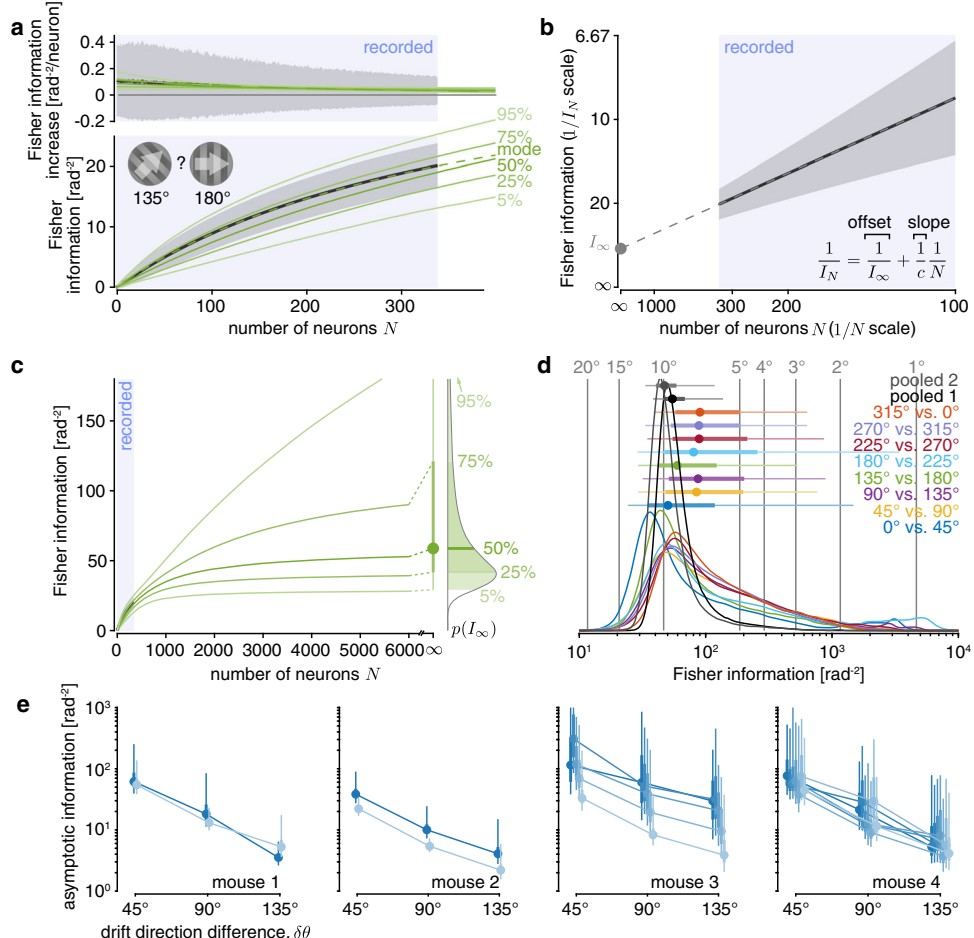

**Fig. 4 Information about drift direction is estimated to asymptote in large neural populations. a** Example information scaling fit, showing data (black; mean estimate ± 1 SD; computed from 135° vs. 180° drift direction trials, as in Fig. 3c) and posterior predictive density for Bayesian fit (green; solid = percentiles, dashed = mode) for the Fisher information increase (top) and Fisher information (bottom) across different population sizes $N$. The model is fitted to the Fisher information increase estimates (top), as these are statistically independent across different population sizes. **b** Plotting the inverse Fisher information $1/I_N$ over the inverse population size $1/N$ (mean estimate ± 1 SD; same data as in **a**) shows an almost perfect linear scaling, as predicted by our theory. Fitting a linear model (gray dashed line) reveals a non-zero asymptotic information $I_\infty$ (gray dot) with $N \to \infty$ **c** The fitted model supports extrapolating the posterior predictive density beyond recorded population sizes (blue shaded area in **a–c**) up to $N \to \infty$. This results in a Bayesian posterior estimate over the asymptotic information $I_\infty$ (right), which we summarize by its median (dot), and its 50% (thick line) and 90% (thin line; truncated at top) credible intervals. **d** Estimates of asymptotic information resulting from different drift direction pairs (colors; $\delta\theta = 45°$ for all pairs) results in comparable posterior densities (colored lines; associated density summaries above densities as in **c**) across different pairs. Therefore, we pooled the data across all non-overlapping pairs with the same $\delta\theta$ to achieve a more precise estimate. The pooled estimates were comparable across two different sets of non-overlapping pairs (gray). The vertical gray lines and numbers indicate the drift direction discrimination thresholds corresponding to different Fisher information estimates. **e** The asymptotic Fisher information estimate (density summaries as in **c**; lines connect posterior medians) is comparable across sessions (different colors; horizontally shifted to ease comparison) and mice.

Comparing these pooled estimates across sessions and mice revealed these estimates to be similar (Fig. 4e). These estimates dropped with an increase in the angular difference $\delta\theta$ in the compared drift directions, as is to be expected from a linear decoder used to discriminate between circular quantities (Supplementary Fig. 10). Together, these observations strongly suggest that the recorded populations were part of a larger population that encoded limited information about the drift direction of the presented stimuli.

**No optimal neural subpopulation across all drift directions.** The recorded population might contain neurons that are not only untuned to drift direction but also do not contribute information through being correlated with other neurons in the population[20,21]. As our information scaling measures are averaged across different orderings of how neurons are added to the population, uninformative neurons would contribute at different population sizes across different orderings. As a result, they make information scaling curves appear shallower than for populations that exclude uninformative neurons. These shallower scaling curves could in turn impact our estimates of asymptotic information (Fig. 4).

To ensure that uninformative neurons did not significantly affect our estimates, we asked if we could identify neural subpopulations within the set of recorded neurons that encode most of the information. Previous work identified such subpopulations in auditory cortex[34] and lateral prefrontal cortex[20] of monkeys, but we are not aware of any work that has shown this for V1. To identify highly informative subpopulations, we ordered the neurons within the recorded population by incrementally adding the neuron that resulted in

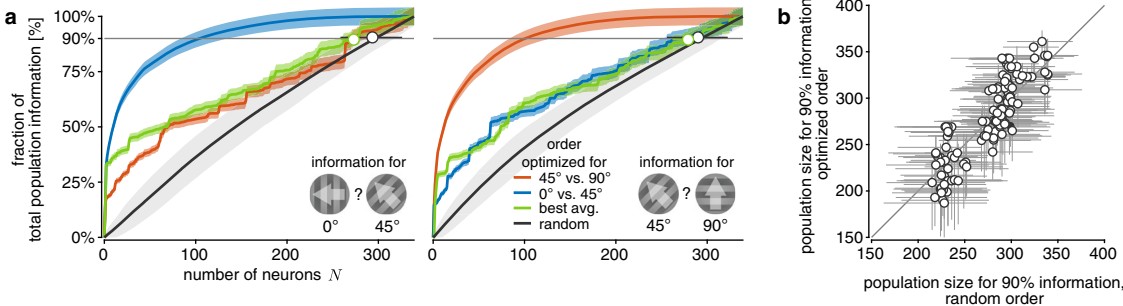

**Fig. 5 No single neural subpopulation appears to encode a disproportionate amount of information across all stimulus drift directions. a** Both panels show that the information increase in the recorded population depends on the order with which neurons are added to the population (colors). The panels differ in the considered drift direction discrimination (left: 0° vs. 45°; right: 45° vs. 90°). The neuron order was optimized by incrementally adding the neuron that resulted in the largest information increase for a 0° vs. 45° (blue) or 45° vs. 90° (orange) drift direction discrimination, or largest average increase across all discriminations with $\delta\theta = 45°$ (green). The optimal ordering for the 0° vs. 45° was also applied to the 45° vs. 90° discrimination (blue line in right panel) and vice versa (orange line in left panel). The average information increase across random orders (black) is shown as baseline reference. Shaded error regions illustrate the uncertainty (mean ± 1 SD) due to limited numbers of trials (all curves), and variability across random orderings (black only). The black and green open circle (bootstrapped median ± 95% CI) show the population sizes required to capture 90% of the information in the recorded population for the associated orderings. **b** Plotting population sizes required to capture 90% of the information in the recorded population (bootstrapped median ± 95% CI) for random ordering vs. orderings optimized to maximize average information across all discriminations revealed no significant difference between the two orderings. Each dot reflects one discrimination for one session.

the largest overall information increase[20,34]. With this ordering, 90% of the information in the recorded population for a particular discrimination could be recovered from only about 30% of the recorded neurons (Fig. 5a). However, natural behavior usually requires information about a wide range of different drift directions rather than the ability to discriminate a specific drift direction pair. To identify how much information the discovered subpopulation contains about other drift directions, we asked how well its population activity supports discriminating another, close-by drift direction pair (Fig. 5a; left vs. right). We found that the same subset of neurons was only able to recover about 55% of the information about this new discrimination. Even a population ordering that boosted the average information across all drift direction pairs did not reveal a highly informative subpopulation within the recorded set of neurons (Fig. 5a; green). To determine whether there is any advantage to a particular ordering, we estimated the population size required to capture 90% of information of the recorded population if we ordered the neurons according to this objective. Across sessions/mice and discriminations, the required population size turns out to not differ significantly compared with a random ordering of the population (Fig. 5b; $t_{63} = -0.215$, two-sided $p \approx 0.83$; across non-overlapping $\delta\theta = 45°$ discriminations). Noise correlations contribute to the observed lack of difference, as this difference becomes significant for trial-shuffled data (Supplementary Fig. 11). If a significant fraction of neurons is uninformative across all drift direction pairs, we would expect these population sizes to differ. Therefore, it is unlikely that our asymptotic information estimates were significantly influenced by the presence of uninformative neurons in the recorded populations.

**Finite-population information impacts asymptotic information.** If estimated asymptotic information mirrors the total information encoded by the animals' brains, it should increase if we increase the amount of information provided by the stimulus in retinal photoreceptor activity. As has been shown previously, higher contrast stimuli result in higher decoding performance from recorded population responses (e.g., see ref. [35]). However, we might observe an information increase in recorded populations even when the asymptotic information remains unchanged (Fig. 6c, right). To determine if increasing the stimulus contrast

results in an increase of asymptotic information, we performed a separate set of experiments in which two mice observed the same drift directions as before, but with a grating contrast of either 10% or 25% that was pseudo-randomly chosen across trials. We hypothesized that the 25% contrast stimuli provide more information about the drift direction, and expected a corresponding increase in asymptotic information.

For most neurons, a contrast increase from 10 to 25% led to a change in baseline activity and re-scaling of their tuning curves, but no appreciable change in pairwise noise correlations (Supplementary Fig. 12). As in correlated populations we cannot predict changes in information solely from changes in tunings, we again moved to measuring information by our generalized Fisher information measure. This revealed that information encoded in the recorded populations significantly increased for higher stimulus contrasts (Fig. 6a for single discrimination and session; Fig. 6b for all sessions/mice, non-overlapping discriminations with $\delta\theta = 45°$: paired $t_{27} = 2.78$, two-sided $p \approx 0.0098$). We in turn applied the same procedure as before (see Fig. 4e) to estimate asymptotic information, but did so separately for the two contrasts (Fig. 6d). We then compared these estimates for $\delta\theta = 45°$ within each session between low- and high-contrast trials (Fig. 6d). In principle, increasing contrast could increase asymptotic information, or it could leave asymptotic information unchanged (Fig. 5c). For three out of the four sessions in which information in the recorded population increased with contrasts for a majority of discriminations (as shown in Fig. 6b), we also observed an increase in asymptotic information with contrast (Fig. 6e, filled dots). This suggests that a more informative stimulus not only increased information in the recorded neural populations but also in the larger (unrecorded) neural population.

**Tens of thousands of neurons decode most of information.** Information in the brain must saturate, as noisy sensors fundamentally limit the sensory information it receives. However, it remains unclear whether information saturates within the population size of V1 (Fig. 1). In our information scaling model, Eq. (1), saturation by definition only occurs in the limit of infinite neurons. We can nonetheless use the model to estimate saturating population sizes by asking how large these populations need to be to encode a large fraction of the asymptotic information (Fig. 7a).

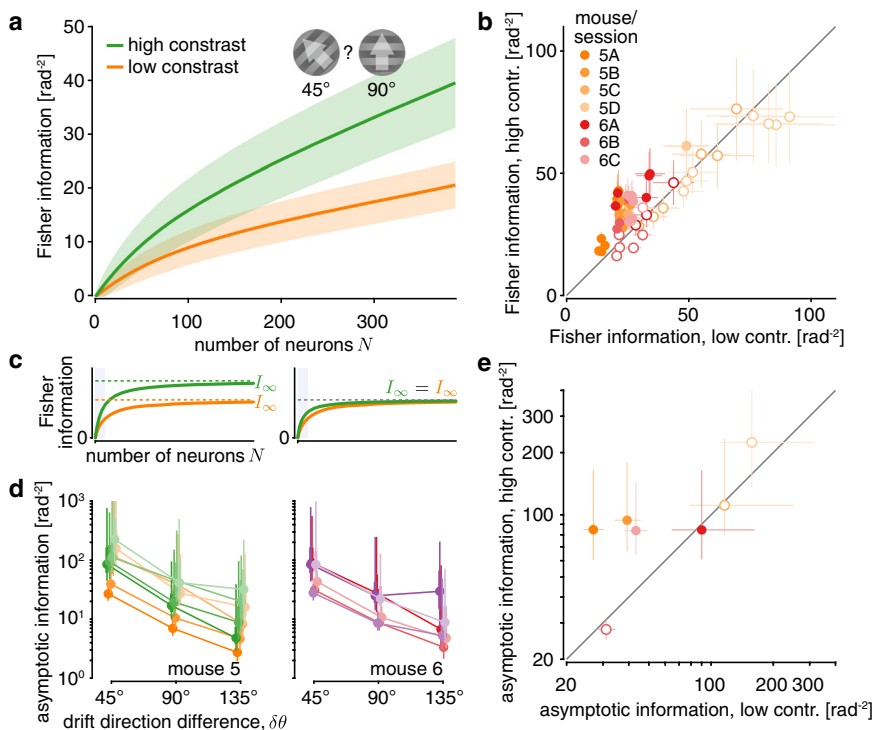

**Fig. 6 Increasing stimulus contrast boosts asymptotic information in V1. a** The information increases more rapidly with population size for high-contrast stimuli (green; mean ± 1 SD) than for low-contrast stimuli (orange; mean ± 1 SD), here shown for the discrimination between 45° vs. 90° drift direction trials of one session of mouse 5. An increase in stimulus contrast significantly increases the total information in the recorded population (bootstrap, two-sided $p < 10^{-5}$). **b** The information in the recorded population was larger for high than low stimulus contrast for most $\delta\theta = 45°$ discriminations across mice and sessions (colors, XY = mouse X, session Y). Each dot shows the information for one discrimination between different drift directions (8 dots per session; error bars = ± 1 SD of the information estimation uncertainty). Filled dots indicate a significant information increase (bootstrap, two-sided $p \geq$ 0.05). **c** Observing an information increase in the recorded population (blue shaded area) does not necessarily imply an increase in asymptotic information (left vs. right). **d** The estimated asymptotic information was generally higher for high-contrast stimuli (green/magenta; shades = sessions) than low-contrast stimuli (orange/red; shades = session; colors as in panel **b**), across different drift direction differences $\delta\theta$ (pooled estimates across different drift direction pairs; posterior density summaries as in Fig. 4c). **e** To compare the pooled asymptotic information estimates for $\delta\theta = 45°$ for low-contrast trials to those for high-contrast trials, we plot them against each other (one dot per session, colors as in panel **b**, error bar centers = posterior medians, error bars = 50% posterior credible intervals). The four filled dots indicate sessions for which the information in the recorded population (panel **b**) is significantly larger for higher contrast stimuli for the majority of discriminations.

We will here focus on population sizes $N_{95}$ that achieve 95% of asymptotic information, which can be found by setting $I_N = 0.95 I_\infty$ in Eq. (1) and solving for $N$. The required population sizes for other fractions of asymptotic information are easily found by a rescaling of $N_{95}$ (Supplementary Fig. 13).

To estimate $N_{95}$, we again relied on the information scaling fits pooled across non-overlapping pairs of drift directions. The recovered population sizes were all on the order of tens of thousands of neurons (Fig. 7b). Our previous analysis (Fig. 5) makes it unlikely that uninformative neurons within the recorded population strongly impact our estimated population sizes. Interestingly, increasing the drift direction difference $\delta\theta$ did not strongly affect these estimates (mice 1–4 in Fig. 7b), even though it modulated asymptotic information (Fig. 4d). Increasing stimulus contrast appeared to increase the estimated population sizes (mice 5–6 in Fig. 7b, orange vs. green), but not consistently so. Thus, it was unclear if a change in information resulted in a global re-scaling of the information scaling curve without changing its shape (Fig. 7c, top), or in the need for more neurons to encode this information (Fig. 7c, bottom).

To clarify the relationship between the asymptotic information $I_\infty$ and required population size $N_{95}$, we did not directly relate these two quantities, as $N_{95}$ is derived from the estimate of $I_\infty$. Instead, we relied on the property that $N_{95}$ is proportional to $I_\infty/c$, where $c$ is the scaling factor associated with the non-limiting

covariance component (see Eq. (1); Methods). Therefore, if $N_{95}$ remains constant across different estimates of $I_\infty$ and $c$, these two quantities need to vary in proportion to each other. In a log–log plot, this implies that the slope describing their relationship would be one. However, we found a slope of $\beta_1 \approx 0.72$, which is slightly, but significantly below one (Fig. 7d; F-test, $F_1 = 21.49$, $p \approx 1.2 \times 10^{-5}$). Substituting the measured relationship between $c$ and $I_\infty$ into the expression for $N_{95}$ results in $N_{95} \approx 4523.8 I_\infty^{0.28}$. This implies that the population size required to encode 95% of the asymptotic information increases with $I_\infty$, but does so only weakly. To illustrate this weak increase, let us consider sessions in which the estimated asymptotic information increased threefold with an increase in stimulus contrast (Fig. 6e). In this case, a population of the size required to capture 95% of the asymptotic information for low-contrast trials could capture 93% of the asymptotic information for high-contrast trials (see "Methods").

**Information is not aligned with principal noise dimensions.** Previous work has observed that most neural population activity fluctuations are constrained to a low-dimensional linear subspace that is embedded in the high-dimensional space of neural activity[36–38]. This might suggest that focusing on such a low-dimensional subspace is sufficient to understand brain function[38]. Thus, we asked if we can recover most of the information about

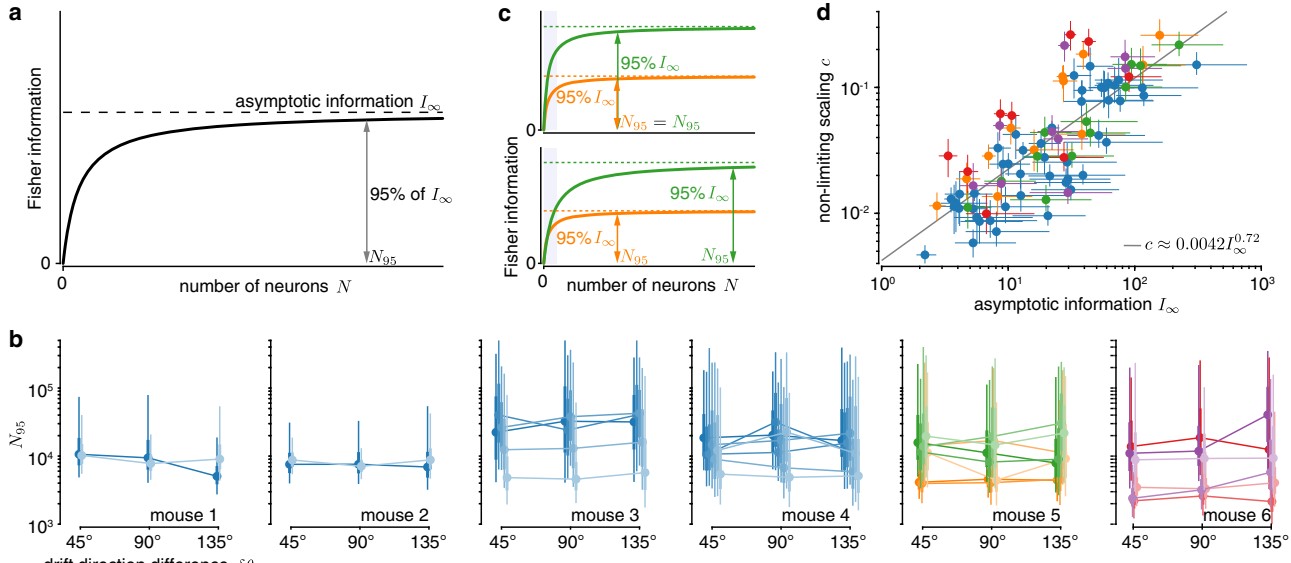

**Fig. 7 Tens of thousands of neurons are required to capture most of the information about stimulus drift direction. a** The TILC predicts how information grows with population size and allows us to estimate the population size $N_{95}$ required to capture 95% of the asymptotic information $I_\infty$. **b** Applied to our data, we find $N_{95}$ in the order of tens of thousands of neurons, consistently across mice (panels) and sessions (colors; blue = uniform contrasts; orange/red = low contrast; green/magenta = high contrast; lines connect individual sessions; horizontally shifted to ease comparison; posterior densities as is Fig. 4c). **c** An increase in information (orange to green) could be achieved by increasing the average information per neuron (top) or by leaving the average information per neuron roughly unchanged while recruiting more neurons (bottom). We would expect $N_{95}$ to grow in the second, but not the first case. These two cases are hard to distinguish from the observed information scaling in smaller populations (shaded blue). **d** Plotting estimated non-limiting scaling $c$ over asymptotic information $I_\infty$ (dot = median, lines = 50% credible interval; colors as in panel **b**) for all animals, sessions, drift direction differences, and contrasts from **b** reveals that $c$ grows sub-linearly with $I_\infty$ (gray line = linear regression of median estimates in log–log plot), which indicates that the estimated population size $N_{95}$ increases weakly with the asymptotic information.

visual drift direction from such subspaces, defined by the dimensions where population activity is most variable. The information encoded in each dimension grows with how well the signal, $\mathbf{f}'$, is aligned with this dimension, but shrinks with the magnitude of noise in this dimension (Fig. 8a; see refs. [17,33]). This tradeoff makes it unclear whether the subspace where population activity is the most variable is indeed the subspace that encodes the most information.

We found the principal dimensions of the noise covariance matrix and asked how much information a subset of the most variable dimensions is able to encode. In our data, 90% of the total variance was captured by approximately 37.6% ± 12.4pp (mean % ± 1 SD percentage points across all sessions/mice, $\delta\theta = 45°$ discriminations) of all available dimensions (Fig. 8b/e), confirming previous reports that relatively few dimensions are required to capture most noise variance. Furthermore, $\mathbf{f}'$ was most strongly aligned to the first few of these principal dimensions[33] (Fig. 8c). Using cosine similarity to measure this alignment, we found that 90% of the cumulative alignment was reached by approximately 7.4% ± 9.1pp of all available dimensions (Fig. 8c/e). Finally, we asked how many dimensions were required to capture 90% of the information encoded in the recorded population. Even though later dimensions were not well-aligned with $\mathbf{f}'$ (see the shallow cumulative alignment increase in Fig. 8c), they were also less noisy (Fig. 8b) and so could contribute significantly to the encoded information. As evident by the continual information growth in Fig. 8d, this resulted in information which was fairly evenly spread across all dimensions, such that, on average, approximately 86.7% ± 2.2pp of all principal noise dimensions were required to encode 90% of all of the recorded information. This is significantly higher than the fraction required to capture 90% of all variance (difference = 48.7 ± 1.5pp, mean ± 1 SEM, paired $t_{63} = 32.53$,

two-sided $p < 10^{-6}$ across non-overlapping discriminations). In fact, if we restricted ourselves to the subspace that captures 90% of all noise variance, we could only decode 58.9% ± 5.6pp of information. Therefore, in our data, relying only on information encoded in the subspace of most variable principal dimensions would result in significant information loss.

## Discussion
We asked how information about the drift direction of a visual stimulus is distributed in large neural populations, and addressed this question by analyzing how information scales with population size. We observed that, in recorded populations, information scaled sublinearly with population size, indicating that noise correlations limited this information. The information scaled in line with TILC if information is indeed limited in larger populations. Based on this theory, we found that we require on the order of tens of thousands of neurons to encode 95% of the asymptotic information. When varying input information by changing stimulus contrast, the required population size appeared to change. Indeed, we found that more information required larger populations, but this relationship was extremely weak. Overall, these findings suggest the presence of information-limiting correlations that cause sensory information in mouse V1 to saturate with population size, indicating the use of a highly redundant, distributed neural code within mouse V1.

Previous attempts at measuring how sensory information scales with population size have frequently found noise correlations to either be beneficial[39] or to not affect information scaling[32,33]. These studies focused on smaller populations (<200 neurons in ref. [39]; <100 neurons in ref. [33]) in which sublinear scaling might be hard to identify (Fig. 1), and in part included spike timing information[39] in addition to the spike counts used

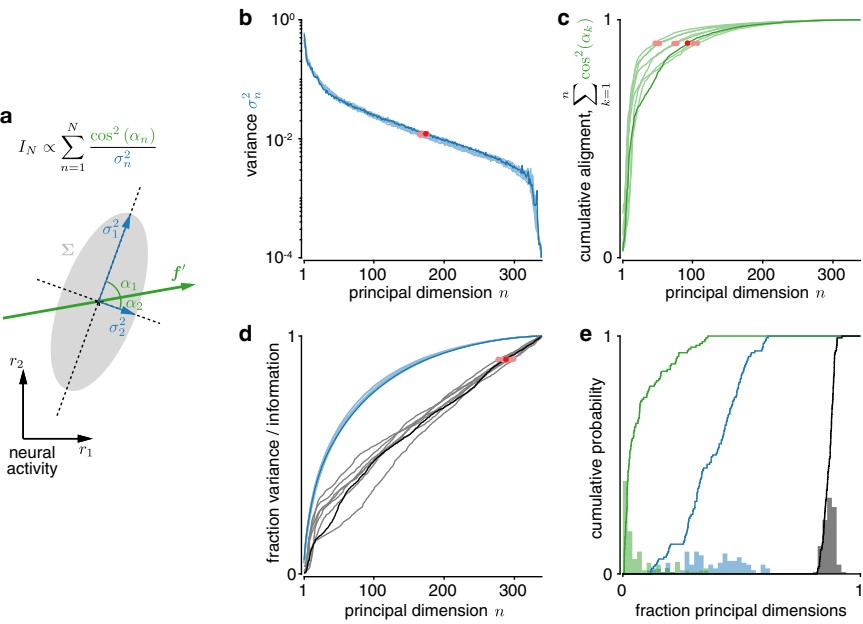

**Fig. 8 Information is not well-aligned with principal noise dimensions. a** The total information in a recorded population, $I_N$, can be decomposed into how well the change in population tuning, $f'$, is aligned to the different principal dimensions of the noise covariance (given by $\cos^2(\alpha_n)$), as well as the noise variances, $\sigma_{n'}^2$, in these dimensions. Low-variance dimensions that are strongly aligned to $f'$ contribute more information. **b** The variance along principal dimensions drops rapidly with dimension (red dots: >90% of total variance). **c** The cumulative alignment of $f'$ to the principal dimensions of the noise covariance rises rapidly with principal dimension (red dots: >90% of total alignment), indicating that $f'$ is most strongly aligned to the first few principal dimensions. **d** The fraction of total information (black; red dots: >90% information) rises more slowly with additional principal dimensions than the fraction of total noise variance (blue). The principal dimensions in panels **b**–**d** are ordered in decreasing order of variance, and show data for the same session as in Fig. 3c (dark = same discrimination as in Fig. 3c; light = other $\delta\theta = 45°$ discriminations for that session). **e** The histograms (bar plots) and cumulative probabilities (lines) of the fractions of the total number of principal dimensions at which the cumulative $f'$ alignment (green), cumulative variance (blue), and cumulative information (black) exceed 90% of their respective totals. These fractions are shown for all $\delta\theta = 45$ discriminations, sessions, and mice. All estimates are cross-validated, and averaged across ten train/test splits (see "Methods").

here. Recent recordings from ~20,000 neurons in mouse V1 suggest information about visual stimuli does saturate[40], but it appears to do so above the population sizes we estimated. These recordings used a slower image scan rate (3 Hz vs. the 30 Hz used for this study), which introduces additional recording noise. This additional noise makes information saturate more slowly with population size (see SI, Sec. 2.3), potentially explaining the larger required population sizes. Recordings from hundreds of neurons in monkey prefrontal cortex revealed sublinear scaling of motor information, compatible with the presence of information-limiting correlations, and resulted in required population size estimates comparable to ours[31]. In contrast to our study, this work measured information about saccade direction rather than about sensory stimulus features. Furthermore, it relied on data from two saccade directions only, and so could not assess if a smaller, selected subpopulation could be used to decode a significant fraction of the total information across a wide range of saccade directions, as we do for drift directions.

Even though information is highly distributed across neurons in a population, most variability is captured by a low-dimensional subspace, leading to suggestions that we might only need to consider the information encoded in this subspace[38]. As we have shown, this argument does not consider that information does not only depend on variability, but also on how the signal aligns with this variability (Fig. 8a). Once both are taken into account, the dimensions of largest variability become a poor proxy for the most informative dimensions (Fig. 8d). This is in line with recent work showing that the most variable subspace in macaque V1 is different from the one that most co-varies between V1 and V2 (ref. [37]), which presumably transmits information between these

areas. Our work explicitly shows such misalignment, and does so in larger populations.

To compare our required population size estimates to the total number of neurons in mouse V1, we conservatively estimated the need for about 48,000 neurons (see "Methods") to achieve drift direction discrimination performance that most likely exceeds that of the animals[23–25]. Our use of time-deconvolved calcium activity as a noisy proxy for spike counts[41,42] makes these estimates upper bounds on required population sizes (see SI). Nonetheless, they compare favorably to the number of neurons in mouse V1, whose estimates range from 283,000 to 655,500 (refs. [43,44]). If we instead compare to the number of neurons in V1 that correspond to the retinotopic area of the visual stimulus, using the entire stimulus or only the full-contrast portion as best and conservative worst-case scenarios, we estimate that the lower and upper bounds on the responsive number of neurons are the same to 10 times higher than our required population size estimates (see "Methods"). This confirms that mouse V1 has more neurons than required to encode most of the estimated asymptotic information about the direction of a moving visual stimulus. Would fewer neurons be required to encode information about natural scenes, which tend to evoke sparser population responses than drifting gratings[45–47]? We do not expect this to be the case, as the fraction of neurons that respond to individual natural stimuli are in fact lower than for drifting gratings, but overall more neurons are required to represent a broad set of natural stimuli[45,47]. This implies that, as for drifting gratings (Fig. 5), we cannot focus on smaller subpopulations that might well discriminate specific image pairs[47], but might fail to convey information about other natural images.

If animals are required to perform tasks that rely on the encoded information we measured (e.g., to discriminate between different drift directions), each neuron in the population would ideally contribute to the animal's choices. Quantified by choice correlations[48,49], an optimal read-out requires the choice correlations of individual neurons to be the fraction of the population's discrimination threshold over that of the neuron[50]. In contrast to previous work (e.g., refs. [51,52]) that found that individual neurons' thresholds match that of the animal, the neurons' average threshold in our data (see information for $N = 1$ in Fig. 3c) is exceedingly small when compared to that of the recorded population (Fig. 3c for full population), and even smaller when compared to estimated asymptotic information (Fig. 4e). This mismatch might arise from shorter stimulus presentation, not tailoring the stimuli to match the neuron's tuning (as done in Britten et al.[51]), recording from lower-level visual areas (V1 vs. V4 or MT) with smaller receptive fields, as well as increased recording noise with calcium imaging as compared to electrophysiological recordings. These lower discrimination thresholds predict increasingly small choice correlations, in line with recent reports from area V1 of monkeys, where fewer than 7% of V1 neurons were found to feature significant choice correlations[53]. In general, the estimated asymptotic information predicted direction discrimination thresholds compatible with previous behavioral reports in mice[23–25], but the use of different stimuli in these experiments precludes a direct quantitative comparison. We furthermore cannot exclude the possibility that mice used a different read-out than the linear one we assumed, or lacked motivation to perform the task to their full potential, further impacting their behavioral performance. A more detailed analysis of the relation between neural activity and choice would require training animals to report their percepts, and then relating these reports to population activity fluctuations.

Multiple factors could have impacted our information measures, and with them our asymptotic information and discrimination threshold estimates. First, the mouse's state of arousal, commonly assessed by their pupil dilation, has been found to fluctuate during similar experiments[28], and such fluctuations could modulate information encoded in V1. Locomotion is linked to arousal[28], and has previously been shown to impact information[29]. In our data, periods of increased locomotion also result in more information in the recorded populations and increase asymptotic information estimates, but do not significantly affect the estimated population sizes required to encode 95% of this asymptotic information (Supplementary Fig. 4). Second, any eye movement within the stimulus presentation period will shift the association between the stimulus and the cells' receptive fields, and result in a relative drop in information. Our stimulus was designed to minimize the effect of eye movements occurring between consecutive stimuli (see "Methods"). Furthermore, eye movement in mice tend to be rare[54] and small[54,55] when compared to the V1 neuron receptive field sizes[56] and size of our stimulus, such that we expect them to have little effect on our estimates of information-limiting correlations. This was confirmed in simulations and theoretical analysis of a simple eye movement model, which revealed that the assumed eye movements might result in over-estimating $N_{95}$, but only in a minor underestimation of $I_\infty$ (Supplementary Fig. 14). Third, we used calcium imaging to obtain dense sampling from large neural populations. Although viral expression of GCaMP6s, as we used here, has been shown to detect nearly all single spikes in some conditions[41], with our imaging conditions, it is likely that we were unable to detect some single spikes. Furthermore, saturation of GCaMP responses might have caused a non-linear mapping between spike counts and measured GCaMP responses, which would quantitatively lower the measured information, but not

qualitatively impact how information scales with population size (Supplementary Fig. 15). Also, neuropil fluorescence has the potential to create shared changes in nearby neurons[57]. We expect that neuropil contamination is unlikely to have a major impact on our information scaling results because such contamination would create redundant signals across neurons and would thus have little impact on information levels that must arise from genuine, non-redundant signals in neurons. However, it is possible that neuropil contamination could have made some uninformative neurons appear informative, in which case a smaller fraction of neurons might be genuinely informative than suggested by Fig. 5. Moreover, residual neuropil fluorescence could cause the non-recorded neuron's signal to "leak out" to recorded neurons, which might result in an underestimation of $N_{95}$. In general, only those factors that modulate information-limiting correlations, which are a small component of the overall noise correlation matrix, impact our information estimates (illustrated in Supplementary Fig. 3). Therefore, while we cannot rule out the presence of such factors, we expect that they did not qualitatively impact our findings.

A prediction of our findings is that neural information should continue to scale according to Eq. (1) in larger populations than those recorded in our experiments. Testing these predictions involves precise estimates of noise correlations, which require about the same number of trials in which the same stimulus (e.g., drift direction) is presented as there are neurons in the population[17,19]. Therefore, even with more powerful recording techniques, information estimates might be limited by the number of trials that can be collected within individual sessions. The use of decoders to estimate information might sidestep these estimates[30,31], with the downside of potentially confounding decoder biases. A further challenge is to record from a population that homogeneously encodes the same amount of information about each stimulus. Such homogeneity ensures that the estimated asymptotic information and population sizes are not specific to particular stimulus values. The weak spatial organization of drift direction selectivity in mouse V1 (ref. [58]) supports this, but the same would be harder to achieve in monkeys due to the much stronger spatial correlations of orientation and direction selectivity in their visual cortices[59]. Finally, even if Eq. (1) is confirmed to match the information in larger populations than used here, it does not allow us to guarantee that the cortex's information is limited by sensory noise and suboptimal computations. Though unlikely, information might continue to grow linearly after an initial sublinear growth[16]. The only way to conclusively rule out this scenario is to record from all neurons in the information-encoding population, which, at least in mammals, will likely not be possible in the foreseeable future[60].

Although all information entering the brain is limited by sensory noise[6], such that it can never grow without bound, the information could be so plentiful or broadly distributed across multiple independent chunks as to not saturate within the population sizes of mammalian sensory areas. In this case, we would expect information to grow on average linearly with the recorded population size, as has been frequently observed in smaller populations. Our findings suggest this not to be the case. However, we suspect the main limiting factor not to be noisy sensors. Instead, most problems that the brain has to deal with require fundamentally intractable computations that need to be approximated, resulting in substantial information loss[61]. Indeed, suboptimal computations can dominate overall information loss, and resulting behavioral variability[62,63], such that they might be the main contributor to the information limitations we observe in our experiments.

If the brain operates in a regime in which information in sensory areas is limited, all information the brain deals with is

uncertain. This idea finds support in the large body of work showing that behavior is well-described by Bayesian decision theory[64–66], which makes effective use of uncertainty. This, in turn, implies that the brain encodes this uncertainty, but its exact neural representations remain unclear[66,67]. A further consequence of limited information is that theories that operate on trial averages (e.g., refs. [68–70]) or assume essentially unlimited information (e.g., ref. [16]) only provide an incomplete picture of the brain's operation. Therefore, an important next step is to refine these theories to account for trial-by-trial variation in the encoded information to achieve a more complete picture of how the brain processes information in individual trials, rather than on average.

## Methods

All experimental procedures were approved by the Harvard Medical School Institutional Animal Care and Use Committee (IACUC).

**Animals and surgery**. Male C57BL/6J mice were obtained from The Jackson Laboratory and housed at 65–75 °F with 35–65% humidity and on a 12-h reverse light/dark cycle. Mice were used for imaging experiments between 4 and 7 months of age. Prior to imaging, mice underwent surgery to implant a chronic cranial window and headplate. Mice were injected intraperitoneally with dexamethasone (3 µg per g body weight) 3–6 h before surgery to reduce brain swelling. During surgery, mice were stably anesthetized with isoflurane (1–2% in air). A titanium headplate was attached to the skull using dental cement (C&B Metabond, Parkell). A ~3.5-mm diameter craniotomy was made over left V1 (stereotaxic coordinates: 2.5 mm lateral, 3.4 mm posterior to bregma). AAV2/1-syn-GCaMP6s (Penn Vector Core) was diluted into phosphate-buffered saline at a final titer of ~2.5E12 gc/ml and mixed 10:1 with 0.5% Fast Green FCF dye (Sigma-Aldrich) for visualization. Virus was injected in a $3 \times 3$ grid with 350 µm spacing near the center of the craniotomy at 250 µm below the dura, with ~75 nl at each site. Injections were made slowly (over 2–5 min) and continuously using beveled glass pipettes and a custom air pressure injection system. The pipette was left in place for an additional 2–5 min after each injection. Following injections, the dura was removed. A glass plug consisting of two 3.5-mm coverslips and one 4.5-mm coverslip (#1 thickness, Warner Instruments) glued together with UV-curable transparent optical adhesive (Norland Optics, NOA 65) was inserted into the craniotomy and cemented in place with cyanoacrylate (Insta-Cure, Bob Smith Industries) and metabond mixed with carbon powder (Sigma-Aldrich) to prevent light contamination from the visual stimulus. An aluminum ring was then cemented on top of the headplate, which interfaced with the objective lens of the microscope through black rubber light shielding to provide additional light-proofing. Data from mouse 1 and 2 were collected as part of a previously published study[71], following a similar surgical protocol. Imaging datasets were collected at least 2 weeks post-surgery, and data collection was discontinued once baseline GCaMP levels and expression in nuclei appeared to be high.

**Visual stimuli**. Visual stimuli were displayed on a gamma-corrected 27-inch IPS LCD gaming monitor (ASUS MG279Q). The monitor was positioned at an angle of 30° relative to the animal and such that the closest point to the mouse's right eye was ~24 cm away, with visual field coverage ~103° in width and ~71° in height. Visual stimuli were generated using PsychoPy[72] or Psychtoolbox (for mice 1 and 2 only) and consisted of square-wave gratings presented on a gray background to match average luminance across stimuli. Gratings were windowed outside of a central circle of radius 20° with a Gaussian of 19° standard deviation, or windowed with a Gaussian central aperture mask of 44° standard deviation (for mice 1 and 2 only) to prevent monitor edge artifacts. Grating drift directions were pseudo-randomly sampled from 45° to 360° in 45° increments at 10 or 25% contrast, spatial frequency of 0.035 cycles per degree, and temporal frequency of 2 Hz. Stimuli were presented for 500 ms, followed by a 500 ms gray stimulus during the inter-stimulus interval (1 Hz presentation). Digital triggers from the computer controlling visual stimuli were recorded simultaneously with the output of the ScanImage frame clock for offline alignment. The visual stimulus was designed to be minimally sensitive to the small eye movements typical of mice[54,55]. In addition to using a full field grating, the stimulus presentation of 500 ms and temporal frequency of 2 Hz was chosen so that each trial consisted of exactly one complete cycle. The effect of fixational eye movements was thus mostly a small shift in phase of the perceived stimulus, which should have little impact on spike counts summed over the full stimulus presentation.

**Microscope design**. Data were collected using a custom-built two-photon microscope. A Ti:Sapphire laser (Coherent Chameleon Vision II) was used to deliver 950 nm excitation light for calcium imaging through a Nikon $16 \times 0.8$ NA water immersion objective, with an average power of ~60–70 mW at the sample. The scan head consisted of a resonant-galvonometric scanning mirror pair separated by a

scan lens-based relay. Collection optics were housed in a light-tight aluminum box to prevent contamination from visual stimuli. Emitted light was filtered (525/50, Semrock) and collected by a GaAsP photomultiplier tube (Hamamatsu). Microscope hardware was controlled by ScanImage 2018 (Vidrio Technologies). Rotation of the spherical treadmill along three axes was monitored by a pair of optical sensors (ADNS-9800) embedded into the treadmill support communicating with a microcontroller (Teensy, 3.1). The treadmill was mounted on an XYZ translation stage (Dover Motion) to position the mouse under the objective.

**Experimental protocol**. Before data acquisition, mice were habituated to handling, head-fixation on a spherical treadmill[73], and visual stimuli for 2–4 days. For each experiment, a field-of-view (FOV) was selected. Multiple experiments conducted in each animal were performed at different locations within V1 or different depths within layer 2/3 (120–180 µm below the brain surface). Before each experiment, the monitor position was adjusted such that a movable flashing stimulus or drifting grating in the center of the screen drove the strongest responses in the imaged FOV, as determined by online observation of neural activity. A single experiment consisted of three blocks of ~45 min each. Once a FOV was chosen, a baseline image (~$680 \times 680$ µm) was stored and used throughout the entire experiment to compare with a live image of the current FOV and manually correct for axial and lateral drift (typically <3 µm between blocks and <10 µm over the full experiment) by adjusting the stage. Drift and image quality stability were verified post hoc by examining 1000 × sped-up movies of the entire experiment after motion correction and temporal downsampling, and experiments that were unstable were discarded without further analysis. Data from mouse 1 and 2 were from previously published experiments[71], where a small fraction of neurons were photostimulated simultaneous to drifting gratings presentation. All photostimulated neurons were excluded from analysis for this paper.

**Data processing**. Imaging frames were first motion-corrected using custom MATLAB code (https://github.com/HarveyLab/Acquisition2P_class) on sub-frame, full-frame, and long (minutes to hours) timescales. Batches of 1000 frames were corrected for rigid translation using subpixel image registration, after which frames were corrected for non-rigid warping on sub-frame timescales using a Lucas-Kanade method. Non-rigid deformation on long timescales was corrected by selecting a global alignment reference image (average of a 1000-frame batch) and aligning other batches by fitting a rigid 2D translation, followed by an affine transform and then nonlinear warping. After motion correction, due to large dataset size (~130 GB), imaging frames were temporally downsampled by a factor of 25 from 30 to 1.2 Hz. Downsampled data were used to find spatial footprints, using a modified version of the constrained nonnegative matrix factorization (CNMF) framework[74] (https://github.com/Selmaan/NMF-Source-Extraction). Three unregularized background components (instead of the default number, one) were used to model spatially and temporally varying neuropil fluorescence, as we observed that the spatial footprints of neuropil activity were distinct from the GCaMP baseline fluorescence background component. We modified the procedure used by CNMF to initialize sources, and instead used an approach to identify sources independently of their spatial profile by using a procedure to cluster pixels based on temporal activity correlations[71]. These sources were then used as initializations for subsequent iterations of the original CNMF algorithm. The resulting spatial footprints from CNMF were used to extract full temporal-resolution fluorescence traces for each source. Traces were deconvolved using the constrained AR-1 OASIS method[75] and individually optimized decay constants. To obtain dF/F, CNMF traces were divided by the average pixel intensity in the absence of neural activity (i.e., the sum of background components and inferred baseline fluorescence from deconvolution of the source's CNMF trace). Because our modified version of CNMF returned sources with both cell-shaped and irregular spatial profiles, we used a convolutional neural network trained on manually annotated labels to classify sources as cell bodies, axial processes (bright spots), horizontal processes, or unclassified. Only data from cell bodies were used in this paper.

To assess neural variability in our recordings, we computed the coefficient of variation (CV; i.e., relative standard deviation) for orientation- and direction-tuned neurons. We found this CV to be roughly one on average, which compares favorably to previously reported mouse V1 data. Bennett et al.[76], for example, found in whole-cell patch clamp recordings a CV of between ~1 (moving) to 2 (stationary) in response to drifting sinusoidal gratings. De Vries et al.[45] found a higher CV of ~2.5 from two-photon calcium imaging data in response to drifting gratings. As fluorescence responses are scaled by some unknown, arbitrary factor relative to spiking activity, we could not compute the neurons' Fano factors. This scaling did not impact our linear Fisher information estimates, as these estimates are invariant to (invertible) linear transformations of neural activity.

**Tuning curve fits**. We used three nested models to fit tuning curves for each neuron. In the direction-tuned model, the average neural response of each neuron was fitted by a mixture of two Von Mises function given by

$$f_1(\theta) = a + b_1 \exp\left(c \cos\left(\theta - \theta_{\text{preferred}}\right)\right)$$
$$+ b_2 \exp\left(-c \cos\left(\theta - \theta_{\text{preferred}}\right)\right), \quad (2)$$

where $a$, $b_1$, $b_2$, $c$, and $\theta_{\text{prefered}}$ are model parameters, and $\theta$ is the stimulus' drift direction. In the orientation-tuned model, the average neural response of each neuron was fitted using a single Von Mises function given by

$$f_2(\theta) = a + b \exp\left(c \cos\left(2\left(\theta - \theta_{\text{preferred}}\right)\right)\right), \qquad (3)$$

with parameters $a$, $b$, $c$, and $\theta_{\text{preferred}}$. The third and last model is a null model that assumes neurons are not significantly tuned to drift direction, and fits a constant value to neural responses, that is $f_3(\theta) = a$. We fitted all three models to the response of neuron across all trials by minimizing the sum of squared residuals between observed neural response and the tuning function across different stimulus drift direction (see Supplementary Fig. 1 for the $R^2$'s associated with these fits). We then compared the nested models by an $F$-test (with Bonferroni correction for multiple comparisons) to test whether neurons are direction-tuned, orientation-tuned or untuned.

**Generalized Fisher information**. Linear Fisher information[17,77,78], which is the Fisher information that can be recovered by a linear decoder, can for stimulus $\theta_0$ be computed by $I(\theta_0) = \mathbf{f}'(\theta_0)^{\mathrm{T}} \Sigma^{-1}(\theta_0) \mathbf{f}'(\theta_0)$. Here, $\mathbf{f}'(\theta_0)$ is the vector of derivatives of each neuron's average response with respect to $\theta$, with the $i$th element given by $\partial f_i(\theta_0)/\partial \theta = \partial \langle r_i | \theta_0 \rangle / \partial \theta$, and $\Sigma(\theta_0) = \text{cov}(\mathbf{r}|\theta_0)$ is the noise covariance of the population activity vector $\mathbf{r}$. Therefore, linear Fisher information is fully determined by the first two moments of the population activity, irrespective of the presence of higher-order moments. Furthermore, if $\hat{\theta} = \mathbf{w}^{\mathrm{T}}(\mathbf{r} - \mathbf{f}(\theta_0)) + \theta_0$ is the unbiased minimum-variance locally linear estimate of $\theta$, its variance is given by $\text{var}\left(\hat{\theta}|\theta_0\right) = 1/I(\theta_0)$[79]. In practice, $\mathbf{f}'(\theta_0)$ and $\Sigma(\theta_0)$ are approximated by their empirical estimates, $\mathbf{f}'(\theta_0) \approx \left(\hat{\mathbf{f}}(\theta_2) - \hat{\mathbf{f}}(\theta_1)\right)/\delta\theta$, and $\Sigma(\theta_0) \approx (\text{cov}(\mathbf{r}|\theta_1) + \text{cov}(\mathbf{r}|\theta_2))$, where $\theta_{1,2} = \theta_0 \mp \delta\theta/2$. This naïve estimate is biased but a bias-corrected estimate can be used[30].

By definition, Fisher information is a measure of fine discrimination performance around a specific reference $\theta_0$, requiring small $\delta\theta$. As we show in the SI, the same measure with $\mathbf{f}'(\theta_0)$ and $\Sigma(\theta_0)$ replaced by their empirical estimate can be used for coarse discrimination for which $\delta\theta$ is larger. Furthermore, this generalization corresponds to $(d'/\delta\theta)^2$, where $d'$ is the sensitivity index used in signal detection theory[22], becomes equivalent to Fisher information in the $\delta\theta \to 0$ limit, and shares many properties with the original Fisher information estimate. In particular, the same bias correction leads to unbiased estimates. Kanitscheider et al.[30] lack an estimate of the variance of the bias-corrected Fisher information estimate that can be computed from data, so we provide a derivation thereof in the SI.

To relate (generalized) Fisher information to discrimination thresholds, we observe that the variance of the stimulus estimate $\hat{\theta}$ is $1/I(\theta_0)$. Assuming this estimate to be Gaussian across trials, the difference in estimates across two stimuli which differ by $\Delta\theta$ is distributed as $N(\Delta\theta, 2/I(\theta_0))$. Therefore, the probability of correctly discriminating these stimuli is $\Phi\left(\Delta\theta\sqrt{I(\theta_0)/2}\right)$[3,80,81], where $\Phi(\cdot)$ is the cumulative function of a standard Gaussian. Setting the desired probability correct to 80% and solving for $\Delta\theta$ results in the drift direction discrimination threshold $\Delta\theta = \Phi^{-1}(0.8)\sqrt{2/I(\theta_0)}$.

**Estimating Fisher information from neural data**. Our Fisher information estimates have two sources of uncertainty. First, they rely on empirical estimates of $\mathbf{f}'(\theta_0)$ and $\Sigma(\theta_0)$ from a limited number of trials that are thus noisy. Second, we assume that recorded neurons to be a small, random subsample of the full population. As we want to estimate the average Fisher information across such subsamples across different population sizes, observing only a single subsample introduces additional uncertainty.

We will first focus on the uncertainty due to a limited number of trials. We can find an unbiased estimate of $I_N$ for a population of $N$ neurons by a biased-corrected estimate $\hat{I}_N$. Our aim is to fit models to how $\hat{I}_N$ changes with $N$. We can estimate this change by computing $\hat{I}_1$ for a single neuron, and then successively add neurons to the population to find $\hat{I}_2, \hat{I}_3, \ldots$ However, this procedure causes $\hat{I}_N$ and $\hat{I}_{N+1}$ to be correlated, as their estimates share the data of the previous $N$ neurons. Therefore, although previous work did not correct for these correlations when fitting the information scaling curves[31–33], it is important to account for them when fitting the information estimates across multiple $N$. Fortunately, the change in information across successive $N$, $\Delta\hat{I}_N = \hat{I}_N - \hat{I}_{N-1}$ is uncorrelated, that is $\text{cov}\left(\Delta\hat{I}_N, \Delta\hat{I}_{N+1}\right) = 0$ (see SI). The intuition underlying this independence is that the response of each neuron can be decomposed into a component that is collinear to the remaining population and one that is independent of it. Only the independent component contributes additional information, making the information increase due to adding this neuron independent of the information encoded in the remaining population. Overall, rather than fitting the information estimates, we will instead fit the information increases across different $N$.

To handle the uncertainty associated with subsampling larger populations, we assumed that the small recorded population is statistically representative of the full population. Then, our aim is to simulate random draws of the size of the recorded population from the full, much larger population. We achieved this simulation by randomly drawing neurons from the recorded population, without replacement, up to the full recorded population size, effectively resulting in a random order of adding recorded neurons to the population. For each such ordering, we estimated the information increase with each additional neuron. As the information in the total recorded population is the same, irrespective of this ordering, the information increases $\Delta I_N$ and $\Delta I_M$ for $N \neq M$ will on average be negatively correlated across different orderings. This is an artifact of re-using the same data to simulate samples from a larger population. As long as the full population is significantly larger than the one we recorded from, the probability of re-sampling the same pair of neurons from the full population is exceedingly small, such that we can ignore these correlations (see SI). Any negative correlations between information increases, however small, will reduce the variance of our Fisher information estimates. Therefore, by ignoring these correlations, we will estimate an upper bound of this variance, and thus overestimate the uncertainty. In summary, we estimated the uncertainty associated with subsampling larger populations by estimating the moments of the Fisher information increase by bootstrap estimates across different orderings with which neurons are added to the population. As shown in Supplementary Fig. 16a, this procedure also captures the uncertainty associated with a limited number of trials, such that no extra correction is needed to account for this second source of uncertainty.

Overall, we estimated the moments of the Fisher information increase $\widehat{\Delta I}_N$ for the discrimination of $\theta_1$ and $\theta_2$ as follows. First, we estimated the empirical moments $\hat{\mathbf{f}}'$ and $\hat{\Sigma}$ using the same number of trials for $\theta_1$ and $\theta_2$. Second, we chose a particular random order with which to add neurons to the population. Third, we used this order to estimate $\widehat{\Delta I}_1, \widehat{\Delta I}_2, \ldots$ by use of the biased-corrected Fisher information estimate applied to $\hat{\mathbf{f}}'$ and $\hat{\Sigma}$. Fourth, we repeated this estimate across $10^4$ different neural ordering to get $10^4$ bootstrap estimates of the Fisher information increase sequence. Fifth, we used the bootstrap estimate to compute the moments $\mu_N = \langle \widehat{\Delta I}_N \rangle$ and $\sigma_N^2 = \text{var}\left(\widehat{\Delta I}_N\right)$ for each $N$, which we in turn use to fit the information scaling curves (see below). As the individual increases are independent across $N$, we used its moments to additionally estimate the moments of $\hat{I}_N = \sum_{n=1}^{N} \widehat{\Delta I}_n$, which are given by $\langle \hat{I}_N \rangle = \sum_{n=1}^{N} \mu_n$ and $\text{var}\left(\hat{I}_N\right) = \sum_{n=1}^{N} \sigma_n^2$. We used these moments to plot the Fisher information estimates in Figs. 3a, 4b/d and 5a.

**Fisher information scaling with limited information**. Moreno-Bote et al.[17] have shown that for large populations encoding limited asymptotic information $I_\infty$, the noise covariance can be decomposed into $\Sigma = \Sigma_0 + I_\infty^{-1}\mathbf{f}'\mathbf{f}'^{\mathrm{T}}$, where only the $\mathbf{f}'\mathbf{f}'^{\mathrm{T}}$ component, called *differential correlations*, limits information. Assuming a population size of $N$ neurons, we can apply the Sherman–Morrison formula to the above noise covariance decomposition[17,50] to find $I_N^{-1} = I_{0,N}^{-1} + I_\infty^{-1}$, where $I_N = \mathbf{f}'^{\mathrm{T}} \Sigma_N^{-1} \mathbf{f}'$ is the Fisher information in this population, and $I_{0,N} = \mathbf{f}'^{\mathrm{T}} \Sigma_0^{-1} \mathbf{f}'$ is the Fisher information associated with the non-limiting noise covariance component $\Sigma_0$. Furthermore, assuming that this non-limiting component contributes average information $c$ per neuron, that is $I_{0,N} = cN$, results in Eq. (1) in the main text. While similar expressions have been suggested before[10,11], they were derived from models that made significantly more restrictive assumptions about neural tuning and shared variability. We also tested a model in which $I_{0,N}$ initially scaled supralinearly in $N$. We found this model by integrating $c(1 - e^{-N/\tau})$ from zero to $N$, resulting in $I_{0,N} = c(N + \tau(e^{-N/\tau} - 1))$ with parameter $\tau$ that controls the extent of the initial supralinearity. The two models become equivalent with $\tau \to 0$. The above derivation relies on the traditional Fisher information definition for fine discrimination. The results remain unchanged when moving to Fisher information generalized to coarse discrimination.

**Fitting information scaling models**. We compared three models for how Fisher information $I_N$ scales with population size $N$. The first *unlim* model assumes linear scaling, $I_N = cN$, and has one parameter, $\phi_1 = \{c\}$. The second *lim* model, given by Eq. (1) in the main text, assumes asymptotic information $I_\infty$, and that the Fisher information associated with the non-limiting covariance component increased linearly, $I_{0,N} = cN$. This model thus has two parameters, $\phi_2 = \{c, I_\infty\}$. The third *lim-exp* model assumes an initial supralinear scaling of $I_{0,N}$, as described above, and has three parameters, $\phi_3 = \{c, I_\infty, \tau\}$. The lim-exp model fits the data consistently worse than the *lim* model (Supplementary Fig. 6b), such we did not consider it in the main text.

As the Fisher information estimates in data are correlated across different population sizes, we did not directly fit these estimates. Instead, we fitted how they changed when adding additional neurons, as the estimated Fisher information increase is uncorrelated across different population sizes. That is, we used the likelihood function $p(X|\phi) = \prod_{n=1}^{N} N\left(\mu_n(X)\big|\Delta I_{n,\phi}, \sigma_n^2(X)\right)$, where $X$ is the recorded data (that is, the recorded population activity in all trials with the drift directions that are being discriminated, yielding the desired moments $\mu_1, \ldots, \mu_N$ and $\sigma_1^2, \ldots, \sigma_N^2$), $\phi$ are the model parameters, $\Delta I_{n,\phi} = I_{n,\phi} - I_{n-1,\phi}$ is the information increase predicted by that model, and $\mu_n$ and $\sigma_n^2$ are the mean and variance of the

estimated information increase in data $X$ for a particular discrimination when moving from population size $n-1$ to $n$ (see further above).

We regularized the fits by weakly informative parameter priors. For $c$ we used $p$ $(c) \propto \mathrm{St}_1(\langle\mu_n\rangle, 100(\langle\mu_n\rangle+0.5)^2)$, which is a Student's t distribution with mean $\langle\mu_n\rangle$, variance $100(\langle\mu_n\rangle+0.5)^2$ and one degree of freedom, and where $\langle\mu_n\rangle$ is the average estimated information increase in the recorded population. Thus, the prior is centered on the empirical estimate for $c$ for the linear scaling model, but has a wide variance around this estimate. We furthermore limited $c$ to the range $c \in [0,\infty]$. For $I_\infty$ we used $p(I_\infty) \propto \mathrm{St}_1\left(\langle\hat{I}_N\rangle, 100 \max\left\{1, \langle\hat{I}_N\rangle\right\}^2\right)$ over $I_\infty \in [0,\infty]$, which is a weak prior centered on the empirical information estimate $\langle\hat{I}_N\rangle = \sum_{n=1}^{N} \mu_n$ for the recorded population. For $\tau$ we used $p(\tau) \propto \mathrm{St}_1(0, N^2)$ over $\tau \in [0,\infty]$. Technically, the data should not inform the priors, as it does here. However, this is not a concern for the extremely weak and uninformative priors used here.

We fitted the different models to data $X$ of individual sessions/mice and discriminations by sampling the associated parameter posteriors, $p(\phi|X) \propto p(X|\phi)p$ $(\phi)$, by slice sampling[82]. The slice sampling interval widths were set to $(\langle\mu_n\rangle+0.5)/2$ for $c$, to $\max\left\{1, \langle\hat{I}_N\rangle\right\}/5$ for $I_\infty$, and to 10 for $\tau$. The samplers were initiated by parameter values found by maximum-likelihood fits for the respective model. For each fit, we sampled four chains with $10^5$ posterior samples each, after discarding 100 burn-in samples, and keeping only each 10th sample. We used the Gelman-Rubin potential scale reduction factor[83] to assess MCMC convergence. To fit the same model to multiple discriminations simultaneously (i.e., our *pooled* fits), we sampled from the pooled posterior $p(\phi|X_{1:K}) \propto p(\phi) \prod_{k=1}^{K} p(X_k|\phi)$, where $X_k$ is the data associated with the $k$th discrimination.

We compared the fit quality of different models by the Watanabe-Akaike information criterion (WAIC; see ref. [84]). This criterion supports comparing models with different numbers of parameters, as it takes the associated change in model complexity into account. It is preferable to the Akaike information criterion or Bayesian information criterion, as it provides a better approximation to the cross-validated predictive density than other methods[85].

We found posterior predictive densities by empirically marginalizing over the posterior parameter samples, $\phi^{(1)},\ldots,\phi^{(J)}$, pooled across all four chains. That is, we approximated the density of any function $f(\phi)$ of these parameters by $p(f|X) \approx J^{-1} \sum_{j=1}^{J} \delta(f - f(\phi^{(j)}))$, where $\delta(\cdot)$ is the Dirac delta function. This approach was used to find the predictive density of the fitted information increase in Fig. 4a (top), as well as the information in Fig. 4a (bottom) and Fig. 4c. We also used it to estimate the posterior distribution of the required population size $N_{95}$ to capture 95% of the asymptotic information.

**Additional data analysis and statistical tests**. Except for Figs. 6 and 7, all statistical tests across sessions/mice were restricted to mice 1–4.

Figure 3. We removed noise correlations in the recorded data by, for each neuron, randomly permuting the trial order across all trials in which the same drift direction was presented. We then compared the total information in the recorded population with ($I_N^{\mathrm{Shuffled}}$) and without ($I_N$) trial-shuffling by a bootstrap test (Fig. 3d). To do so, we estimated mean and variance of that total recorded information as described above, and then computed the probability of the null hypotheses ($I_N^{\mathrm{Shuffled}} \leq I_N$) by $p = \mathrm{pr}(I_N^{\mathrm{Shuffled}} - I_N < 0)$, where we assumed Gaussian information estimates. We compared $I_N^{\mathrm{Shuffled}}$ to $I_N$ across sessions/mice by a paired $t$-test across all non-overlapping discriminations with $\delta\theta = 45°$ (Fig. 3d). We focused exclusively on discriminations that did not share any drift directions, to avoid comparing estimates that rely on the same underlying set of trials. Unless otherwise noted, all non-overlapping discriminations with $\delta\theta = 45°$ were performed on the 0° vs. 45°, 90° vs. 135°, 180° vs. 225°, and 270° vs. 315° discriminations. To test for significant differences in the drift direction discrimination thresholds (Fig. 3f) across multiple discriminations with the same difference in drift directions, $\theta$, we relied on the one-to-one mapping between information and discrimination threshold, and performed the test directly on the estimated information. For $K$ discriminations (in our case $K = 4$ for non-overlapping discriminations), let $I_{N,k}$, $k = 1,\ldots,K$ denote the information in the recorded population for discrimination $k$, $I_{N,k} \sim \mathrm{N}\left(\mu_{N,k}, \sigma_{N,K}^2\right)$. To test the null hypothesis that all $I_{N,k}$ share the same mean, we drew $10^5$ bootstrap samples each from $TS_{H_1} = \sum_{k=1}^{K}\left(I_{N,k} - \mu_{N,k}\right)^2$ and $TS_{H_0} = \sum_{k=1}^{K}\left(I_{N,k} - \mu_N\right)^2$ with $\mu_N = K^{-1}\sum_{k=1}^{K} \mu_{N,k}$, and then computed the probability that $TS_{H_0}$ is larger than $TS_{H1}$ by $p = \mathrm{pr}(TS_{H_1} - TS_{H_0} < 0)$.

Figure 4. To test how $1/I_N$ scales with $1/N$ (Fig. 4b), we found the moments of $1/I_N$ by $\langle 1/I_N\rangle \approx 1/\langle I_N\rangle$ and $\mathrm{var}(1/I_N) \approx \mathrm{var}(I_N)/I_N^4$. To fit $\langle 1/I_N\rangle$ over $1/N$, we performed weighted linear regression with weights $1/\mathrm{var}(1/I_N)$ for each $N$. The pooling across different discriminations in Fig. 4d was performed over 45° vs. 90°, 135° vs. 180°, 225° vs. 270°, and 0° vs. 315° for pooled 1, and 0° vs. 45°, 90° vs. 135°, 180° vs. 225°, and 270° vs. 315° for pooled 2. All other pooled estimates (Figs. 4e, 6d and e, and 7b) were pooled across 45° vs. 90°, 135° vs. 180°, 225° vs. 270°, and 0° vs. 315° for $\delta\theta = 45°$, across 45° vs. 135°, 90° vs. 180°, 225° vs. 315°, and 0° vs. 270° for $\delta\theta = 90°$, and across 45° vs. 180°, 90° vs. 315°, and 0° vs. 225° for $\delta\theta = 135°$. Note that the estimate $I_N$'s are correlated across different $N$'s, and we did not correct for

these correlations. Such a correction might lower the reported $R^2$ values. Therefore, the Bayesian model comparison across different information scaling models, as reported in the main text, provides a statistically sounder confirmation of limited asymptotic information.

Figure 5. The shaded error regions in Fig. 5a relied on parametric bootstrap estimates. For information scaling for a fixed ordering, we computed the estimate and variance of $I_1, I_2,\ldots$ by the Fisher information and the variance of this estimator (see SI), and used these estimates to compute mean and variance of the information increase associated with adding individual neurons to the population. We then re-sampled these information increases from Gaussian distributions with the found moments, and summed the individual samples to find different samples for the whole information scaling curve. These samples were in turn used to estimate mean and variance of the information scaling for a fixed order with which neurons were added to the population. This procedure was chosen, as the increase in Fisher information is independent across added neurons, whereas the total Fisher information is not. A similar procedure was used to find the estimates for random orderings, for which we additionally shuffled the order of neurons across different samples of the information scaling curve. The above procedures yielded $10^3$ bootstrap samples for each information scaling curve, which we in turn used to find samples for the population sizes required to capture 90% of the total information (Fig. 5a, b). In neither case did we apply bias correction of the Fisher information estimate. This bias correction would have been stronger for larger population sizes, which would have led to a seeming (but not real) drop of information with population size, resulting from a lower number of trials per neuron in the population, and an associated stronger bias correction.

Figure 6. To identify for individual discriminations if increasing the stimulus contrast increased information in the recorded population (Fig. 6a, b), we estimated information in the recorded population by the bias-corrected Fisher information estimate[30], and its variance by our analytical expression for this estimate's variance (see SI). We assumed the estimate for low and high contrast, $I_N^{\mathrm{LO}}$ and $I_N^{\mathrm{HI}}$, to be Gaussian, and found the probability of no information increase by $\mathrm{pr}(I_N^{\mathrm{HI}} \leq I_N^{\mathrm{LO}})$, using the aforementioned moments. The paired $t$-test across sessions/mice (Fig. 6b) did not take into account the information estimates' variance. For Fig. 6e, higher contrast was considered to significantly increase the information in the recorded population (filled dots in Fig. 6e), if it did so for at least five out of eight possible discriminations with $\delta\theta = 45°$.

Figure 7. To test the relationship between $c$ and $I_\infty$ in Fig. 7d, we performed the linear regression $log_{10}(c) = \beta_0 + \beta_1 log_{10}(I_\infty)$. The relationship between $N_{95}$ and $I_\infty$ was found by substituting $c = 10^{\beta_0} I_\infty^{\beta_1}$ into the expression for $N_{95}$, resulting in $N_{95} = 0.95 I_\infty^{1-\beta_1}/(0.05 \times 10^{\beta_0})$. To find the information loss for using a smaller population size than required, we assumed $I_\infty^{\mathrm{hi}} = \alpha I_\infty^{\mathrm{lo}}$ and computed the fraction $I_N^{\mathrm{hi}}/I_\infty^{\mathrm{hi}}$ at $N = N_{95}^{\mathrm{lo}}$, which is the population size that captures 95% of $I_\infty^{\mathrm{lo}}$. Substituting the found relationships between $I_\infty$, $c$, and $N_{95}$ results in this fraction to be given by $0.95/(0.95 + 0.05\alpha^{1-\beta_1})$, which, for $\alpha = 3$, equals 0.93. Interestingly, this fraction depends only the relationship between $I_\infty^{\mathrm{lo}}$ and $I_\infty^{\mathrm{hi}}$, as quantified by $\alpha$, but not on their individual values.

Figure 8. All estimates in Fig. 8 are averages across 10 random splits of the recorded data. For each split, half of the trials were used to compute the principal dimensions, $\mathbf{Q}_{\mathrm{train}}$, using the spectral decomposition $\mathbf{\Sigma}_{\mathrm{train}} = \mathbf{Q}_{\mathrm{train}} \mathbf{D}_{\mathrm{train}} \mathbf{Q}_{\mathrm{train}}^{\mathrm{T}}$, where $\mathbf{D}_{\mathrm{train}}$ is diagonal, $\mathbf{Q}_{\mathrm{train}}$ is the matrix of unit eigenvectors, and we denote the $n$th column vector of $\mathbf{Q}_{\mathrm{train}}$ by $\mathbf{q}_{n,\mathrm{train}}$. The second half of trials was used to find $\mathbf{f}'_{\mathrm{test}}$ and $\mathbf{\Sigma}_{\mathrm{test}}$, from which we computed the shown estimates as follows. The noise variance associated with the $n$th principal dimension was found by $\mathbf{q}_{n,\mathrm{train}}^{\mathrm{T}} \mathbf{\Sigma}_{\mathrm{test}} \mathbf{q}_{n,\mathrm{train}}$. The $\mathbf{f}'$ alignment to the $n$th principal dimension was found by $\cos^2(\alpha_n) = \left(\mathbf{q}_{n,\mathrm{train}}^{\mathrm{T}} \mathbf{f}'_{\mathrm{test}}\right)^2/\mathbf{f}'_{\mathrm{test}}{}^{\mathrm{T}} \mathbf{f}'_{\mathrm{test}}$. The information encoded in the first $n$ principal dimensions was found by $I_n = \mathbf{f}'_{\mathrm{test}}{}^{\mathrm{T}} \mathbf{Q}_{1:n,\mathrm{train}} \left(\mathbf{Q}_{1:n,\mathrm{train}}^{\mathrm{T}} \mathbf{\Sigma}_{\mathrm{test}} \mathbf{Q}_{1:n,\mathrm{train}}\right)^{-1} \mathbf{Q}_{1:n,\mathrm{train}}^{\mathrm{T}} \mathbf{f}'_{\mathrm{test}}$, where $\mathbf{Q}_{1:n,\mathrm{train}}$ is the matrix formed by the first $n$ columns of $\mathbf{Q}_{\mathrm{train}}$.

*Additional analyses in discussion*. To compare the estimated population sizes to the number of neurons in V1, we asked for the number of neurons required to encode 95% of the asymptotic information associated with a direction discrimination threshold of 1°. This threshold most likely exceeds the behavioral performance that mice can reach even for high contrast stimuli[23,25] and thus provides an upper bound on the required population size. Achieving such a low threshold requires an asymptotic information of 4651 rad$^{-2}$ (Fig. 3e), and approximately 48,000 neurons are necessary to encode 95% of this information (Fig. 7d). Current estimates of the neural density of mouse V1 range from 92,400 to 214,000 neurons per mm$^3$ (refs. [43,44]). For area V1 with an approximate size of 3.063 mm$^3$ (ref. [43]), this amounts to 283,000 to 655,500 neurons[44]. Therefore, our estimated population sizes are well within those available in V1 of mice. In addition to comparing our estimates to the total number of neurons in V1, we also considered best and worst-case scenarios for the number of neurons in V1 that correspond to the retinotopic area of the visual stimulus (103° azimuth, 71° elevation). To convert between degrees of visual space and mm of cortical space, we used the conversion factors 63°/mm in azimuth and 40°/mm in elevation[86]. In the best-case scenario, the entire visual stimulus corresponds to ~1.65 × 1.78 mm, or 2.95 mm$^2$ in the cortex. Relative to the total area of V1, estimated as ~3.25–4 mm$^2$ (refs. [87,88]), 75–90% of V1

neurons would be activated by the stimulus. Using the range above for total neurons in V1, this is on the order of ~10× our estimates for the number of neurons encoding 95% of asymptotic information. For a conservative worst-case scenario, we consider only the full-contrast portion of the stimulus (circle with radius 20°), for which the retinotopic area covered is ~0.5 mm², or ~12.5–15% of V1 neurons. This conservative estimate of a lower bound on the number of responsive neurons is ~1× our required population size estimates. Thus, mouse V1 has more neurons than required to encode most of the estimated asymptotic information about the direction of a moving visual stimulus.

**Reporting summary**. Further information on research design is available in the Nature Research Reporting Summary linked to this article.

## Data availability
The datasets generated and analyzed during this study are available in the Figshare repository, https://doi.org/10.6084/m9.figshare.13274951. Source data are provided with this paper.

## Code availability
MATLAB code performing the described analyzes and generating the resulting figures is available at https://doi.org/10.5281/zenodo.4291863.

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

## Acknowledgements

We would like to thank Alexandre Pouget, Peter Latham, and members of the HMS Neurobiology Department for useful discussions and feedback on the work, and Rachel Wilson and Richard Born for comments on early versions of the manuscript. The work was supported by a scholar award from the James S. McDonnell Foundation (grant# 220020462 to J.D.), grants from the NIH (R01MH115554 to J.D.; R01MH107620 to C.D. H.; R01NS089521 to C.D.H.; R01NS108410 to C.D.H.; F31EY031562 to A.W.J.), the NSF's NeuroNex program (DBI-1707398. to R.N.), MINECO (Spain; BFU2017-85936-P to R.M.-B.), the Howard Hughes Medical Institute (HHMI, ref 55008742 to R.M.-B.), the ICREA Academia (2016 to R.M.-B.), the Government of Aragon (Spain; ISAAC lab, cod T33 17D to I.A.-R.), the Spanish Ministry of Economy and Competitiveness (TIN2016-80347-R to I.A.-R.), the Gatsby Charitable Foundation (to R.N.), and an NSF Graduate Research Fellowship (to A.W.J.).

## Author contributions

All authors designed the research and wrote the paper; A.W.J. and S.N.C. performed the experiments; M.K., R.N., I.A.-R., R.M.-B., and J.D. developed the theory; and M.K., A.W.J., S.N.C., and J.D. analyzed the data.

## Competing interests

The authors declare no competing interests.
