## [Peer Review File · Nature Communications]

Reviewers' comments:

Reviewer #1 (Remarks to the Author):

The manuscript addresses a long-standing question in systems neuroscience – the degree to which shared variability in neuronal responses limits their information capacity. Historically, addressing this question has been challenging as most experimental datasets have been limited to small numbers of simultaneously recorded neurons. Pairwise noise correlations, which can be estimated from such datasets, are not sufficient to answer this question. Therefore, the systematic analysis of information content of a large simultaneously recorded neuronal populations presented in this manuscript is very timely.

The manuscript makes several novel contributions. The authors generalize linear Fisher information to coarse discriminations with relatively weak assumptions about the distributions of neuronal population responses. The authors then use this measure to show that linear Fisher information saturates as the number of neurons increases, demonstrating the presence of information limiting correlations in population responses. By extrapolating from the recorded population sizes, the authors estimate that 95% of the asymptotic information would be contained in 10s of thousands of neurons. Overall, the analyses in the paper are rigorously executed, and the conclusions are generally compelling and well-supported by data. I believe the concerns listed below could be readily addressed by qualifying some of the conclusions or in the discussion.

(i) My only major concern arises from the technical challenge of reliably estimating spiking based one two-photon calcium imaging data. A somewhat surprising conclusion of the paper is that no ordering appears to be significantly better at capturing 90% of total population information than random ordering, suggesting that uninformative neurons are rare in the recorded population. However, demonstrating this definitively based on calcium imaging data is challenging due to contaminating neuropil signals. Although tuning of V1 neurons is heterogeneous, neuropil fluorescence nevertheless shows modulation by orientation and may carry additional information in the form of noise correlations (see e.g. <https://dx.doi.org/10.3389%2Fncir.2017.00050>). Therefore, residual neuropil fluorescence may make entirely unresponsive neurons appear informative in the analysis in Figure 5. I think the conclusions of this section should be qualified and additional information should be provided on the procedure applied for neuropil correction in the Methods (see minor point (1) below).

(ii) When considering the number of neurons needed to capture 95% of asymptotic information, it might make more sense to compare it to the number of neurons in the part of V1 activated by the grating rather than the entirety of V1. Taking this into account, N95 is probably not that different from the number of neurons available.

(iii) I would have also liked to see some discussion of the applicability of the authors' results to more natural stimulation conditions. Drifting gratings used in this study tend to activate a relatively large fraction of V1 neurons (about half in the present dataset), while natural images excite V1 more sparsely.

Minor issues:

1) The description of pre-processing steps used to extract activity traces is unclear. The methods specify that spatial footprints were identified "using a CNMF-based method". The linked repository appears to contain a fork of CNMF code from Pnevmatikakis et al. so I assume CNMF was used as described by the original authors. Please clarify. It is also unclear whether CNMF was used to simply define the spatial footprints – this is my reading of lines 722-724 – or whether CNMF temporal components were used. If the former is the case, please provide more information as to how activity traces were extracted and how neuropil correction was applied. Finally, please include a citation to Pnevmatikakis et al. for the CNMF algorithm.

2) Please provide the standard deviation of the Gaussian aperture mask used for grating presentation.

Typos:

Supplementary information, page 12: last line should read "easier than alternative model", rather than "easier that alternative models".

Supplementary information, page 17: first line "to satisfies" should be "to satisfy".

Reviewer #2 (Remarks to the Author):

Kafashan et al. study the encoding of movement direction in mouse visual cortex, by using calcium imaging to record from large populations of neurons while the mouse views drifting gratings. Decoding the movement direction from the neural signals, they inferred how much stimulus information was present in neural populations of different sizes. These analyses enabled them to show that the information saturates with increasing population size, and to estimate the size of neural population needed to capture 95% of the available stimulus information. The saturation of information is consistent with the neural population having information-limiting correlations (as analyzed by Moreno-Bote et al. 2014), and that theoretical framework was used to assess the total amount of available stimulus information (e.g., the asymptotic amount found by extrapolating to infinite population size).

I think this study is thorough, well-executed, and informative. My enthusiasm is slightly diminished by the following factors, that could perhaps be addressed by the authors in a revision:

- (As alluded to by the authors) The data processing inequality tells us that, with finite stimulus information at the retina, cortical stimulus information must saturate with increasing population size, so long as the cortical population is sufficiently larger than the retinal one such that the cortical information capacity is larger than the retinal one (and it is). And the Moreno-Bote 2014 paper showed us that saturating information required information-limiting correlations. So I don't think we should be surprised that we observe saturating information, consistent with information-limiting correlations, in cortex.

- It is of course good to test these theoretical ideas experimentally, but that test was recently done, in monkey, by Bartolo et al.

I think that the paper would be strengthened if the authors could address the above critique, and highlight substantial advances that I've missed in my assessment -- e.g., beyond that the experiments were done in mouse not monkey, contained some advances in the analysis methods, and contained an estimate of the N95.

My more minor suggestions are as follows:

1) I'm curious about whether the animal's behavior matters. It's known that running behavior modulates V1 firing rates (e.g., Neill and Stryker Neuron 2010), and that changing firing rates changes correlations between neurons (e.g., de La Rocha et al. Nature 2007). As such, I'm left wondering if the information-limiting correlations could depend on the behavior. This isn't a critical thing to add for this paper, but would be accessible within the authors' dataset, and could be quite interesting, e.g., if running reduced information-limiting correlations and raised the information in V1.

2) Related to point (1) is the open question of whether attention effects (which change correlations between neurons) would alter the information-limiting correlations. If those correlations are reduced in the presence of attention, then the info saturation level could be quite different when

the animal is engaging their visual system vs passively viewing the stimuli. This is especially relevant given the discussion (e.g., line 627) relating behavioral performance to V1 information coding.

3) I found the plotting convention in Fig. 4B to be a bit confusing. I understand that the authors want to plot the inverses against each other, and to have a horizontal axis where increasing number of neurons goes to the right. But then to have the vertical axis have increasing information going down (also plotting inverse information), makes it needlessly confusing. I don't think it's critical, but I would have found this easier to parse if they would have just plotted $1/\text{info}$ vs $1/N$ and not flipped the axes around.

4) Deconvolving calcium imaging to infer neural spiking is error-prone (e.g., the Ledochowitsch 2019 paper). The authors discuss a bit that imaging noise means that their information estimates are lower bounds (which I agree with). I wonder, though, if there could be more going on, that could warrant some consideration. Given the calcium imaging is biased towards prolonged depolarization events -- including multi-spike bursts -- the activity events detected by Ca^{++} imaging likely contain fewer single spike events than are present in the true underlying neural activity. If that's right, then the results in this paper could pertain more to the "multi-spike-event neural code for movement direction". That code could be different from the "isolated spike neural code for movement direction" (e.g., multiplexing ideas from Harvey, et al., ... Bensmaia, 2013 PLoS Biology).

Reviewer #3 (Remarks to the Author):

The submitted article titled "Scaling of sensory information in large neural populations reveals signatures of information-limiting correlations" by Kafashan et al. is the latest in the area of research that examines how a population of neurons encodes sensory information. This area of neural coding really took off with Shadlen & Newsome's modeling paper in 1998 and several studies throughout the early-to-mid-2000's in the cat and macaque using several different methods and drawing several different conclusions. At that time, researchers were only recording from several dozen neurons and looking at even smaller populations because of limitations of the methods and numbers of trials. Depending on the task, the number of neurons, whether timing is important, and whether spatiotemporal coding is important can all lead to different conclusions about whether a population of neurons encode more information as a whole (synergistic or cooperative), are mostly independent, or encode less information as a whole compared to the sum of the parts (redundant). Correlated variability, tuning heterogeneity, and how correlated variability is related to the stimulus all play an important factor in these measurements. What the current submission offers is a look at these questions in a larger population of neurons (several hundred) in the mouse, and they even go further by extrapolating their analysis out to 10's of thousands of neurons. They conclude that correlated variability causes V1 neurons to be redundant, but despite this, a reasonably-sized population of neurons can encode all the necessary information about the direction of drift of a grating.

Major Concerns:

Extrapolating data is a dangerous endeavor in even ideal conditions. My primary concern is that this paper offers me no confidence that the data they start with was collected in ideal conditions. Even in the original Shadlen & Newsome paper, they include commentary about how experimental conditions might contribute to correlated variability. These conditions are external and not true neural correlated variability. This is variability introduced by the experiment itself and it will generally be correlated across neurons. To truly measure the influence of correlated variability, every trial should occur under the same conditions. This is a highly important and sometimes underappreciated problem in neuroscience. There are several factors related to this that are not sufficiently addressed in this paper:

1. There is virtually no raw data presented in the paper to even attempt to alleviate my concern. There are no single trial responses shown, no variability or fano factor measurements, no raw tuning curves, or anything else related to this question presented. We get one image and a bunch of tuning curve fits.

2. Were eye position and eye movements measured? Any change in the stimulus-receptive field relationship will increase variability and retinotopy will cause it to be correlated.

3. Many V1 responses are highly variable depending on running behavior. The methods say that running speed and direction were available, but I did not see it presented or discussed anywhere. Just by watching GCaMP responses in real-time, running causes massive changes in responses.

4. The behavioral state of the animal matters, but these mice were not doing any task and nothing was mentioned about pupil size or anything else that might signal changes in motivation or alertness, etc. The authors do discuss this factor related to the primate literature (Marlene Cohen's work), but never in the context of how it really affects their estimates of redundancy and extrapolation. It is not only that attention reduces correlated variability, which is what was discussed. The mouse could be changing from "attentive" to "non-attentive" from trial-to-trial.

5. Orientation-dependent adaptation and offset responses will lead to substantial trial-to-trial variability. When randomly interleaving trials, you introduce variability because the response will be in a different state for every trial purely based on the previous stimulus presented. This is especially concerning because trials and inter-trials were only 500 ms in duration. Offset responses are almost certainly going to run into the next trial. This is especially true with GCaMP. I have seen offset responses last over 2 seconds even with static stimuli let along drifting gratings. The effects of adaptation last for several seconds. The orientation biases in mouse will likely cause this variability to be correlated across neurons as well.

6. I am also concerned about using such low contrast stimuli for mice. Many times tuning curves measured at 10% just look like a bunch of noise compared to tuning curves measured in the same neurons with 50% contrast. If you combine a very weak stimulus drive with all the factors above, correlated variability is going to dominate the responses. What are the R-squared measurements for your tuning curve fits? Are tuning curves consistent between 10 and 25%? What does Fig. 3 look like with the 25% contrast data?

7. I would also like to see Fig. 3 replotted with all of the manipulations done in Fig. 5. This leads me to another overall general major concern. This relates to the whole premise of the paper. Is there ever anywhere in the brain that is looking at all of V1 at the same time? I would think that population coding (whether it is pooling or whatever) would always be from the perspective of particular subgroups. So why not look at what shuffling does to the optimal group of neurons for doing just one task? I would think that would be more relevant to what the brain is likely to implement if the mouse were to actually perform the task. The ideal group of neurons would presumably develop more highly weighted connections to sensorimotor areas related to the task through training. For example, 0-degree tuned neurons would be linked to "lick left" and 45-degree tuned neurons would be linked to "lick right". I would think that the neurons unrelated to the discrimination task are unlikely to be used for decoding, since they will not even respond to the stimulus. If these neurons have no information about the stimulus, all they are contributing is correlated variability. Specific subgroups of neurons seems more related to mouse behavioral tasks and would be consistent with Newsome-style micro-stimulation results as well. If there is something I am missing, this needs to be discussed in more detail.

Minor Concerns:

1. Using drift direction instead of orientation can be a little confusing. Generally, orientation tuning

and direction selectivity are the two descriptions of gratings, where the latter is always referring to 180-degree differences in direction. With random dot noise, direction of motion is not linked to the orientation of the stimulus.

2. Fig 5a, left (0-degree grating has wrong orientation)

3. There is a paragraph labeled discussion in the methods that seems to be out of place?

Response to reviewers

We would like to thank the reviewers for the insightful and constructive comments. We have revised the manuscript to address all the points raised, and hope that the reviewers appreciate these changes.

In particular, we now provide more details on single-neuron responses, as shown in the new Figs. S1 and S10, and details on the pairwise noise correlations, as shown in the new Figs. S2 and S10. This has revealed pairwise correlations magnitudes in line with those previously observed (Cohen & Kohn, 2011), as well as that the tuning curves in response to 25% contrast stimuli are for most neurons shifted and scaled versions of the tuning curves in response to 10% contrast stimuli. Furthermore, we performed additional analyses on how running speed - as a proxy for the animals' arousal - impacts information and information scaling (see new Fig. S3) that revealed that running boosts population (as previously observed) and asymptotic information, but not the number of neurons required to encode 95% of asymptotic information. Last, but not least, we have revised sections of the manuscript to explain in more detail the difference of our work to Bartolo et al. (2020), to discuss in more detail the various experimental factors that might have impacted our information estimates, and to elaborate in more detail on the design of our visual stimulus as well as how we preprocessed our calcium trace images.

Please find below a detailed point-by-point response to the reviewers' individual comments. The comments are *italics* and our responses are in normal text.

Reviewer #1

The manuscript addresses a long-standing question in systems neuroscience – the degree to which shared variability in neuronal responses limits their information capacity. Historically, addressing this question has been challenging as most experimental datasets have been limited to small numbers of simultaneously recorded neurons. Pairwise noise correlations, which can be estimated from such datasets, are not sufficient to answer this question. Therefore, the systematic analysis of information content of a large simultaneously recorded neuronal populations presented in this manuscript is very timely.

The manuscript makes several novel contributions. The authors generalize linear Fischer information to coarse discriminations with relatively weak assumptions about the distributions of neuronal population responses. The authors then use this measure to show that linear Fischer information saturates as the number of neurons increases, demonstrating the presence of information limiting correlations in population responses. By extrapolating from the recorded population sizes, the authors estimate that 95% of the asymptotic information would be contained in 10s of thousands of neurons. Overall, the analyses in the paper are rigorously executed, and the conclusions are generally compelling and well-supported by data. I believe the concerns listed below could be readily addressed by qualifying some of the conclusions or in the discussion.

We would like to thank the reviewer for their fair, rigorous and positive assessment of our work, and are glad that they appreciated the significance of our findings and contributions, and our analyses' rigour.

(i) My only major concern arises from the technical challenge of reliably estimating spiking based one two-photon calcium imaging data. A somewhat surprising conclusion of the paper is that no ordering appears to be significantly better at capturing 90% of total population information than random ordering, suggesting that uninformative neurons are rare in the recorded population. However, demonstrating this definitively based on calcium imaging data is challenging due to contaminating neuropil signals. Although tuning of V1 neurons is heterogeneous, neuropil fluorescence nevertheless shows modulation by orientation and may carry additional information in the form of noise correlations (see e.g. <https://dx.doi.org/10.3389%2Fncir.2017.00050>). Therefore, residual neuropil fluorescence may make entirely unresponsive neurons appear informative in the analysis in Figure 5. I think the conclusions of this section should be qualified and additional information should be provided on the procedure applied for neuropil correction in the Methods (see minor point (1) below).

We thank the reviewer for raising this point. We do not think neuropil contamination contributes significantly to the main information scaling results because signal from contamination will be shared (correlated) among many neurons. This would create redundant information across multiple neurons and thus would not increase the information in the population. For example, if in a pool of neurons, another neuron is added that intrinsically has no signal but has neuropil contamination, the additional neuron will not help decode the stimulus, because whatever signal was present in the neuropil contamination is likely already in the population. This is an advantage of our approach, compared to the reference the reviewer cites which examines pairwise noise correlations. We therefore think neuropil contamination is unlikely to contribute in a major way to most of the analyses in the manuscript.

We agree that neuropil contamination has the potential to make uninformative neurons appear informative. As mentioned above, this is not an issue if the real signal is already captured in the population, but it could be a problem if that real signal is not present. The main place of concern, as the reviewer notes, is in the random ordering of neurons. Because we cannot completely rule out neuropil contamination in this specific analysis, we have added a sentence to the Discussion that states:

“Also, neuropil fluorescence has the potential to create shared changes in nearby neurons (Lee, Meyer, Park, & Smirnakis, 2017). We expect that neuropil contamination is unlikely to have a major impact on our information scaling results because such contamination would create redundant signals across neurons and would thus have little impact on information levels that must arise from genuine, non-redundant signals in neurons. However, it is possible that neuropil contamination could have made some uninformative neurons appear informative, in which case a smaller fraction of neurons might be genuinely informative than suggested by Figure 5.”

(ii) When considering the number of neurons needed to capture 95% of asymptotic information, it might make more sense to compare it to the number of neurons in the part of V1 activated by the grating rather than the entirety of V1. Taking this into account, N95 is probably not that different from the number of neurons available.

We thank the reviewer for bringing up this point. To determine the number of neurons in V1 activated by the grating, we estimated the best and worst-case scenarios for neurons that correspond to the retinotopic area covered by either the full stimulus or only a central portion, which we then compare to the required population size to capture 95% of asymptotic information. Using this approach, we estimate that the range of neurons is ~1x-10x greater than the N95. We have now added these comparisons to the Discussion:

“If we instead compare to the number of neurons in V1 that correspond to the retinotopic area of the visual stimulus, using the entire stimulus or only the full-contrast portion as best and conservative worst-case scenarios, we estimate that the lower and upper bounds on the responsive number of neurons are the same to 10 times higher than our required population size estimates (see Methods).”

We have also elaborated upon this in the Methods:

“In addition to comparing our estimates to the total number of neurons in V1, we also considered best and worst-case scenarios for the number of neurons in V1 that correspond to the retinotopic area of the visual stimulus (103° azimuth, 71° elevation). To convert between degrees of visual space and mm of cortical space, we used the conversion factors 63°/mm in azimuth and 40°/mm in elevation (Kalatsky & Stryker, 2003). In the best-case scenario, the entire visual stimulus corresponds to ~1.65 x 1.78mm, or 2.95mm² in cortex. Relative to the total area of V1, estimated as ~3.25-4mm² (Garrett, Nauhaus, Marshel, & Callaway, 2014; Waters et al., 2019), 75-90% of V1 neurons would be activated by the stimulus. Using the range above for total neurons in V1, this is on the order of ~10x our estimates for the number of neurons encoding 95% of asymptotic information. For a conservative worst-case scenario, we consider only the full-contrast portion of the stimulus (circle with radius 20°), for which the retinotopic area covered is ~0.5mm², or ~12.5-15% of V1 neurons. This conservative estimate of a lower bound on the number of responsive neurons is ~1x our required population size estimates. Thus, mouse V1 has more neurons than required to encode most of the estimated asymptotic information about the direction of a moving visual stimulus.”

(iii) I would have also liked to see some discussion of the applicability of the authors' results to more natural stimulation conditions. Drifting gratings used in this study tend to activate a relatively large fraction of V1 neurons (about half in the present dataset), while natural images excite V1 more sparsely.

This is a good point that we didn't consider in the previous version of the manuscript. What is important here is to differentiate between population sparsity, i.e., a low number of neurons responding to individual stimuli, and lifetime sparsity, i.e., a low number of neurons responding across all stimuli. Population responses are indeed more sparse for natural images than for drifting gratings, as has been shown in cat (Baddeley et al., 1997), monkey (Vinje & Gallant, 2000), and mouse V1 (de Vries et al., 2019; Yoshida & Ohki, 2020). This stands in contrast to the number of neurons that are involved in responding across all stimuli, which - at least in mouse V1 - appears higher for natural stimuli (de Vries et al., 2019; Yoshida & Ohki, 2020). Therefore, fewer neurons would be required to discriminate between individual natural images (as also shown in Yoshida & Ohki, 2020), but downstream areas would need to "listen" to more neurons to capture the information encoded across a large range of natural images. In the context of our analysis, we would expect to see a more rapid rise of information in Fig. 5a for the discrimination of image pairs, but not for discriminating across a large range of natural images. As a consequence, we would not expect our N95 estimates to change.

To make this clear in the manuscript, we have added the following to the Discussion:

"Would fewer neurons be required to encode information about natural scenes, which tend to evoke sparser population responses than drifting gratings (de Vries et al., 2020; Vinje & Gallant, 2000; Yoshida & Ohki, 2020)? We don't expect this to be the case, as the fraction of neurons that respond to individual natural stimuli are in fact lower than for drifting gratings, but overall more neurons are required to represent a broad set of natural stimuli (de Vries et al., 2020; Yoshida & Ohki, 2020). This implies that, as for drifting gratings (Fig. 5), we cannot focus on smaller subpopulations that might well discriminate specific image pairs (Yoshida & Ohki, 2020), but might fail to convey information about other natural images."

Minor issues:

1) *The description of pre-processing steps used to extract activity traces is unclear. The methods specify that spatial footprints were identified "using a CNMF-based method". The linked repository appears to contain a fork of CNMF code from Pnevmatikakis et al. so I assume CNMF was used as described by the original authors. Please clarify. It is also unclear whether CNMF was used to simply define the spatial footprints – this is my reading of lines 722-724 – or whether CNMF temporal components were used. If the former is the case, please provide more information as to how activity traces were extracted and how neuropil correction was applied. Finally, please include a citation to Pnevmatikakis et al. for the CNMF algorithm.*

We have added more details on the pre-processing steps (see below). We have also added citations to Pnevmatikakis et al. for CNMF and Friedrich et al. for deconvolution. In brief, we modified the initialization routine for input to the core CNMF algorithm to be less restrictive about source shapes, and then used those sources to initialize the original CNMF.

“[...] Downsampled data was used to find spatial footprints, using a modified version of the constrained nonnegative matrix factorization (CNMF) framework (Pnevmatikakis et al., 2016) (<https://github.com/Selmaan/NMF-Source-Extraction>). Three unregularized background components (instead of the default number, one) were used to model spatially and temporally varying neuropil fluorescence, as we observed that the spatial footprints of neuropil activity were distinct from the GCaMP baseline fluorescence background component. We modified the procedure used by CNMF to initialize sources, and instead used an approach to identify sources independently of their spatial profile by using a procedure to cluster pixels based on temporal activity correlations (Chettih & Harvey, 2019). These sources were then used as initializations for subsequent iterations of the original CNMF algorithm. The resulting spatial footprints from CNMF were used to extract full temporal-resolution fluorescence traces for each source. Traces were deconvolved using the constrained AR-1 OASIS method (Friedrich, Zhou, & Paninski, 2017) and individually-optimized decay constants. To obtain dF/F, CNMF traces were divided by the average pixel intensity in the absence of neural activity (i.e., the sum of background components and inferred baseline fluorescence from deconvolution of the source’s CNMF trace). Because our modified version of CNMF returned sources with both cell-shaped and irregular spatial profiles, we used a convolutional neural network trained on manually annotated labels to classify sources as cell bodies, axial processes (bright spots), horizontal processes, or unclassified. Only data from cell bodies were used in this paper.”

2) Please provide the standard deviation of the Gaussian aperture mask used for grating presentation.

Gratings were windowed with a Gaussian mask of 19 degrees standard deviation outside of a central circle of radius 20 degrees, or windowed with a central Gaussian aperture mask of 44 degrees standard deviation (mice 1 and 2 only). We apologize for this omission and have updated the Methods accordingly.

Typos:

Supplementary information, page 12: last line should read “easier than alternative model”, rather than “easier than alternative models”.

Supplementary information, page 17: first line “to satisfies” should be “to satisfy”.

Thank you, fixed.

Reviewer #2

Kafashan et al. study the encoding of movement direction in mouse visual cortex, by using calcium imaging to record from large populations of neurons while the mouse views drifting gratings. Decoding the movement direction from the neural signals, they inferred how much stimulus information was present in neural populations of different sizes. These analyses enabled them to show that the information saturates with increasing population size, and to estimate the size of neural population needed to capture 95% of the available stimulus information. The saturation of information is consistent with the neural population having information-limiting correlations (as analyzed by Moreno-Bote et al. 2014), and that theoretical framework was used to assess the total amount of available stimulus information (e.g., the asymptotic amount found by extrapolating to infinite population size).

I think this study is thorough, well-executed, and informative.

We would like to thank the reviewer for their appreciation of our work, and the constructive comments.

My enthusiasm is slightly diminished by the following factors, that could perhaps be addressed by the authors in a revision:

- (As alluded to by the authors) The data processing inequality tells us that, with finite stimulus information at the retina, cortical stimulus information must saturate with increasing population size, so long as the cortical population is sufficiently larger than the retinal one such that the cortical information capacity is larger than the retinal one (and it is). And the Moreno-Bote 2014 paper showed us that saturating information required information-limiting correlations. So I don't think we should be surprised that we observe saturating information, consistent with information-limiting correlations, in cortex.

This is an important point - thank you for bringing it up. Indeed, it is clear from the data processing inequality that information in cortex cannot be larger than information that enters through the retina. Therefore, information needs to saturate at some point. What remained unclear is the point at which it does. One could imagine that the information is so plentiful (as suggested by Ecker et al., 2011), or represented in such a distributed way, that information saturation is not apparent in the recorded neurons, even when recording from large populations - even the whole of V1. It is also exactly what has plagued previous work that tried to identify such saturation: they did not find such saturation in recorded populations, and so could not tell if

the brain operates in a regime in which information saturates *within the number of neurons available to the brain*. Our data suggests that it does. Furthermore, we took the extra care when extrapolating to larger numbers of neurons as it is critical for such extrapolations to make statistically sound predictions about the further growth of information, and the population sizes involved. We judge these contributions as sufficiently novel to make them of interest to a large readership.

We acknowledge, however, that even though we briefly touched on the above in the Discussion in the previous version of the manuscript, it might not have been sufficiently clear. Therefore, we have now modified the relevant section to read

“Although all information entering the brain is limited by sensory noise (Faisal et al., 2008), such that it can never grow without bound, the information could be so plentiful or broadly distributed across multiple independent chunks as to not saturate within the population sizes of mammalian sensory areas. In this case, we would expect information to grow on average linearly with the recorded population size, as has been frequently observed in smaller populations. Our findings suggest this not to be the case. [...]”

- It is of course good to test these theoretical ideas experimentally, but that test was recently done, in monkey, by Bartolo et al.

We agree that Bartolo et al. made important contributions to addressing some of the questions we are also asking, and we have already commented in the previous version about the differences between ours and their results. We realize, however, that they were not clear enough, so following the reviewer’s request we have further emphasized the differences.

We would like to start by saying that the most important difference between Bartolo et al. and our results is that they ask the question of information about a motor variable, while we address the question of information about sensory variables. So our paper presents a - to our knowledge - first demonstration of information-limiting correlation in the sensory domain, where the question of limited information was originally posed.

Second, the information the brain needs to process in natural scenarios is usually not about telling apart two specific saccade or drift directions, but information about saccade or drift direction in general. Even though we measure information by discriminating between pairs of drift direction pairs, we do so across multiple such pairs in order to identify information about drift direction in general. Bartolo et al. couldn’t do this, as they only recorded neural data for a single pair of saccade directions (left vs. right).

In addition, there are a number of other methodological differences. The most important one is that Bartolo et al. extrapolate information by assuming that information estimates for different population sizes are independent. As we have pointed in our paper (in main text, Methods, and SI), this assumption is incorrect, and might lead to misleading estimates. We thus have developed a new fitting procedure to avoid this problem.

We have clarified the above points in the main text by rewriting the paragraph in Section “Neural signature of limited asymptotic information”:

“These estimates are correlated across different population sizes, as estimates for larger populations share data with estimates for smaller populations. Unlike previous work that estimated how information scales with population size (Bartolo, Saunders, Mitz, & Averbek, 2020; Cotton et al., 2018; Mendels & Shamir, 2018), we accounted for these correlations by fitting how information increases with each additional neuron, rather than fitting the total information for each population size. This information increase turns out to be statistically independent across population sizes (see Methods), making the fits statistically sound and side-stepping the problem of fitting correlated data.”

In the Discussion, we have rewritten the paragraph

“Recordings from hundreds of neurons in monkey prefrontal cortex revealed sublinear scaling of motor information, compatible with the presence of information-limiting correlations, and resulted in required population size estimates comparable to ours (Bartolo et al., 2020). In contrast to our study, this work measured information about saccade direction rather than about sensory stimulus features. Furthermore, it relied on data from two saccade directions only, and so could not assess if a smaller, selected subpopulation could be used to decode a significant fraction of the total information across a wide range of saccade directions, as we do for drift directions.”

I think that the paper would be strengthened if the authors could address the above critique, and highlight substantial advances that I've missed in my assessment -- e.g., beyond that the experiments were done in mouse not monkey, contained some advances in the analysis methods, and contained an estimate of the N95.

We hope that the above points make sufficiently clear the novelties of our study, and how those differ from the recent findings of Bartolo et al.

My more minor suggestions are as follows:

1) I'm curious about whether the animal's behavior matters. It's known that running behavior modulates VI firing rates (e.g., Neill and Stryker Neuron 2010), and that changing firing rates changes correlations between neurons (e.g., de La Rocha et al. Nature 2007). As such, I'm left wondering if the information-limiting correlations could depend on the behavior. This isn't a critical thing to add for this paper, but would be accessible within the authors' dataset, and could be quite interesting, e.g., if running reduced information-limiting correlations and raised the information in VI.

Thank you for this very relevant suggestion. In light of previous work, it is indeed interesting to see how information and our associated asymptotic estimates are affected by running speed. Furthermore, as increased running speed is frequently interpreted as indicating a heightened state of attention or arousal (see below), it might also give us a window into how information is modulated by attention.

As suggested, we now replicated some of the previous analysis while splitting the data into trials with low and high running speed, and are showing the results in a new Fig. S3. As expected from previous work (e.g., Dardalot & Stryker, 2017), we found that a higher running speed resulted in increased information in the recorded population. Furthermore, we found that it also significantly increased our asymptotic information estimates. Interestingly, we didn't find the same statistically significant increase in the required population size to encode 95% of this asymptotic information.

In addition to the new Fig. S3, we describe its result in multiple locations throughout the text. In Section "Noise correlations limit information" we added:

"Indeed, higher running speeds, which were previously used as a proxy for increased attention (McGinley et al., 2015), resulted in increased information (as shown previously by (Dardalot & Stryker, 2017)) and lower thresholds (Fig. S3)."

In the Discussion we added:

"First, the mouse's state of arousal, commonly assessed by their pupil dilation, has been found to fluctuate during similar experiments (McGinley et al., 2015), and such fluctuations could modulate information encoded in V1. Locomotion is linked to arousal (McGinley et al., 2015), and has previously been shown to impact information (Dardalot & Stryker, 2017). In our data, periods of increased locomotion also result in more information in the recorded populations and increase asymptotic information estimates, but do not significantly affect the estimated population sizes required to encode 95% of this asymptotic information (Fig. S3)."

2) Related to point (1) is the open question of whether attention effects (which change correlations between neurons) would alter the information-limiting correlations. If those correlations are reduced in the presence of attention, then the info saturation level could be quite different when the animal is engaging their visual system vs passively viewing the stimuli. This is especially relevant given the discussion (e.g., line 627) relating behavioral performance to V1 information coding.

This is indeed an interesting question. We did not control for attention in our experiments, and so unfortunately cannot analyze its impact on neural activity and information. However, if we assume that increased attention is reflected in increased locomotion (as is commonly done, e.g., Mineault et al., 2016), then our above analysis applies. This analysis has shown that

information increases with running speed, and so, presumably, also with increased attention, which is consistent with a significant body of previous work on how attention increases information. We hope that the additions to the Discussion (see above) will alert the reader of a potential modulation of information by attention that might also exist in our dataset and that could be taken as a future direction of research in more controlled experiments.

3) I found the plotting convention in Fig. 4B to be a bit confusing. I understand that the authors want to plot the inverses against each other, and to have a horizontal axis where increasing number of neurons goes to the right. But then to have the vertical axis have increasing information going down (also plotting inverse information), makes it needlessly confusing. I don't think it's critical, but I would have found this easier to parse if they would have just plotted $1/\text{info}$ vs $1/N$ and not flipped the axes around.

We agree with the reviewer that reversing the horizontal axis might be unnecessarily confusing. In the revised manuscript, Fig. 4b shows $1/\text{info}$ over $1/N$ without the flipped axis.

4) Deconvolving calcium imaging to infer neural spiking is error-prone (e.g., the Ledochowitsch 2019 paper). The authors discuss a bit that imaging noise means that their information estimates are lower bounds (which I agree with). I wonder, though, if there could be more going on, that could warrant some consideration. Given the calcium imaging is biased towards prolonged depolarization events -- including multi-spike bursts -- the activity events detected by Ca^{++} imaging likely contain fewer single spike events than are present in the true underlying neural activity. If that's right, then the results in this paper could pertain more to the "multi-spike-event neural code for movement direction". That code could be different from the "isolated spike neural code for movement direction" (e.g., multiplexing ideas from Harvey, et al., ... Bensmaia, 2013 PLoS Biology).

We agree that calcium imaging is unlikely to detect all single spike events. For this reason, we used the experimental approaches best suited to capturing as many action potentials as possible. Specifically, we used viral expression of GCaMP6s, which in our hands creates a much higher signal-to-noise ratio than the transgenic GCaMP mice we have tested. In fact, we started this project using transgenic mice, but for concerns of spike detection and aberrant activity in those mice, we moved to using entirely viral expression. Recent papers comparing calcium imaging and electrophysiology, including the one cited by the reviewer, have focused on spike detection in transgenic mice. To our knowledge, the best characterization of spike detection with GCaMP6s expression from a virus is the original paper (Chen et al. Nature 2013). That paper showed ~99% single-spike detection rates (see their Fig. 3f, black bar). Also, a recent pre-print has directly noted the worse spike detection with transgenic mice than with viruses (<https://www.biorxiv.org/content/10.1101/840686v1>). However, it is true that spike detection rates can vary greatly with imaging conditions and exact expression levels. We have therefore now added a paragraph to the Discussion that includes potential caveats. In this paragraph we include a sentence related to this point:

“Third, we used calcium imaging to obtain dense sampling from large neural populations. Although viral expression of GCaMP6s, as we used here, has been shown to detect nearly all single spikes in some conditions (T.-W. Chen et al., 2013), with our imaging conditions, it is likely that we were unable to detect some single spikes.”

Reviewer #3

The submitted article titled “Scaling of sensory information in large neural populations reveals signatures of information-limiting correlations” by Kafashan et al. is the latest in the area of research that examines how a population of neurons encodes sensory information. This area of neural coding really took off with Shadlen & Newsome’s modeling paper in 1998 and several studies throughout the early-to-mid-2000’s in the cat and macaque using several different methods and drawing several different conclusions. At that time, researchers were only recording from several dozen neurons and looking at even smaller populations because of limitations of the methods and numbers of trials. Depending on the task, the number of neurons, whether timing is important, and whether spatiotemporal coding is important can all lead to different conclusions about whether a population of neurons encode more information as a whole (synergistic or cooperative), are mostly independent, or encode less information as a whole compared to the sum of the parts (redundant). Correlated variability, tuning heterogeneity, and how correlated variability is related to the stimulus all play an important factor in these measurements. What the current submission offers is a look at these questions in a larger population of neurons (several hundred) in the mouse, and they even go further by extrapolating their analysis out to 10’s of thousands of neurons. They conclude that correlated variability causes V1 neurons to be redundant, but despite this, a reasonably-sized population of neurons can encode all the necessary information about the direction of drift of a grating.

We would like to thank the reviewer for the detailed assessment of our work, and the constructive comments.

Major Concerns:

Extrapolating data is a dangerous endeavor in even ideal conditions. My primary concern is that this paper offers me no confidence that the data they start with was collected in ideal conditions. Even in the original Shadlen & Newsome paper, they include commentary about how experimental conditions might contribute to correlated variability. These conditions are external and not true neural correlated variability. This is variability introduced by the experiment itself and it will generally be correlated across neurons. To truly measure the influence of correlated variability, every trial should occur under the same conditions. This is a highly important and sometimes underappreciated problem in neuroscience. There are several factors related to this that are not sufficiently addressed in this paper:

We agree that extrapolation needs to be performed with utmost care. For this reason we hope that the reviewer appreciates the rigor we have put into the methodology that underlies our

extrapolations - in particular the explicit assessment of the involved uncertainties (see, for instance, our response to Reviewer 2's comments).

We also agree that careful experimental design is necessary to avoid spurious findings. Related to this, we would like to emphasize that only a specific type of noise correlation structure impacts our information measures - namely those *differential* correlations that introduce variability in the direction of f (i.e., the change in mean population activity with stimulus). These correlations are only one small component of the overall noise correlation matrix. Therefore, a global rise in the noise correlations due to some potentially uncontrolled factors, such as microsaccades, eye blinks, or fluctuations of attention, does *not* necessarily imply that this rise impacted our information measures. Furthermore, it is experimentally impossible to ensure that every trial occurs under the same condition - even anesthetized animals feature up/down states that cannot be controlled for experimentally, and might impact correlations and information. Therefore, we made sure to critically assess all factors that might impact our results, and added additional statements and analyses to the manuscript about their potential impact. We have provided further details about this below. We hope that these address the reviewer's concerns.

1. There is virtually no raw data presented in the paper to even attempt to alleviate my concern. There are no single trial responses shown, no variability or fano factor measurements, no raw tuning curves, or anything else related to this question presented. We get one image and a bunch of tuning curve fits.

To address the reviewer's concerns, we have now added a new Fig. S1 that includes more examples for per-trial neural responses, raw tuning curves, associated tuning curve fits, and R^2 values for these fits. In another new Fig. S2, we furthermore provide pair-wise noise correlation measures, which are in line with those previously found (Cohen & Kohn, 2011). The reason why we haven't provided much information about individual tuning curve fits is that those were only used to classify the neurons as direction-tuned, orientation-tuned or untuned. Neither the fits nor the classification was used in any of the further analysis that is central to the manuscript (we have, in fact, been previously criticized for presenting these fits, as they were considered misleading for exactly that reason). We understand that some readers might nonetheless want to see additional details on the responses of individual neurons, which we now provide in the newly added figure. Furthermore, we have updated the manuscript by adding the following sentences to the first paragraph in Results:

“See Figs. S1 and S2 for more examples of neural responses, tuning curves, and pairwise noise correlations. [...] Tuning curves were plotted for the sole purpose of characterizing individual neural responses, but our fits had no bearing on any of our further analysis.”

We now point the readers to the R^2 values in Methods where we describe how we fitted the tuning curves:

“We fitted all three models to the response of neuron across all trials by minimizing the sum of squared residuals between observed neural response and the tuning function across different stimulus drift direction (see Fig. S1 for the R^2 's associated with these fits).”

The recorded dF/F relate to neural activity by an unknown scaling factor. This unknown scaling factor makes it impossible to compute Fano factors, which is why we do not provide them for our data. Not knowing this scaling factor was not an issue when computing information, as Fisher information (and our generalized variant) is invariant under invertible linear transformations, even if these transformations are unknown.

2. Were eye position and eye movements measured? Any change in the stimulus-receptive field relationship will increase variability and retinotopy will cause it to be correlated.

We did not measure eye position and movement during the experiments, and so cannot be certain that eye movements did not impact our results. Nonetheless, we expect any impact of eye movement to be minimal, because

- eye movements are generally rare in mice, ranging from 0.12 to 0.3 saccades/sec (Keller et al., 2012), such that they should have only impacted a small number of trials in our experiments, and
- eye movements in mice are small and have been reported to be on the order of ~2-5 degrees (Keller et al. 2012, Ayaz et al. 2013), which is smaller than both the average receptive field size of V1 neurons (5-10 degrees, Niell & Stryker 2008) and the visual space covered by our stimulus.
- Furthermore, our stimulus was designed such that small differences in eye position across trials should have little impact on the information provided to the animal. Specifically, the duration of the stimulus presentation and temporal frequency of the grating were chosen so that precisely one full cycle of the grating was shown every trial. On a coarse level, this would mean that no matter how eye position drifts, the neuron is still exposed to one full grating cycle. The phase might be shifted, depending on fixation drift. However, given that we summed neural activity throughout the full trial into a single number, this concern might be minimal.
- Translational eye movements are expected to generate neuronal correlations with little or no projection onto the axis where stimulus drift directions can be better discriminated (a rotational rather than translational discrimination), and are thus expected to have little to no effect on our estimates of information-limiting correlations.

We now acknowledge in the Discussion that eye position and movement might have impacted our information estimates:

“Second, any eye movement within the stimulus presentation period will shift the association between the stimulus and the cells' receptive fields, and result in a relative drop in information. Our stimulus was designed to minimize the effect of

eye movements occurring between consecutive stimuli (see Methods). Furthermore, eye movement in mice tend to be rare (G. B. Keller, Bonhoeffer, & Hübener, 2012) and small (Ayaz, Saleem, Schölvinck, & Carandini, 2013; G. B. Keller et al., 2012) when compared to the V1 neuron receptive field sizes (Niell & Stryker, 2008) and size of our stimulus, such that we expect them to have little effect on our estimates of information-limiting correlations.”

Furthermore, we added more details about the stimulus design to Methods:

“The visual stimulus was designed to be minimally sensitive to the small eye movements typical of mice (Ayaz et al., 2013; G. B. Keller et al., 2012). In addition to using a full field grating, the stimulus presentation of 500ms and temporal frequency of 2Hz was chosen so that each trial consisted of exactly one complete cycle. The effect of fixational eye movements was thus mostly a small shift in phase of the perceived stimulus, which should have little impact on spike counts summed over the full stimulus presentation.”

3. Many V1 responses are highly variable depending on running behavior. The methods say that running speed and direction were available, but I did not see it presented or discussed anywhere. Just by watching GCaMP responses in real-time, running causes massive changes in responses.

Indeed, as has been previously shown, locomotion in mice affects both population activity (e.g., Mineault et al., 2016), as well as stimulus information (e.g., Dadarlat & Stryker, 2017). We have now performed a similar analysis, and confirm that higher running speeds result in more information in the encoded population, as well as in our asymptotic information estimates (Fig. S3). It did not significantly impact the estimated population sizes to encode 95% of asymptotic information, though.

Averaging across different running speeds yields lower information estimates when compared to periods of rapid running. However, even for the latter, our saturation information growth model fits the data better than the non-saturating one. Therefore, while averaging across different running speeds might have quantitatively impacted our results, we don't expect it to qualitatively change our findings.

We have now added information about our running speed analysis to the Section ““Noise correlations limit information” we added:

“Indeed, higher running speeds, which were previously used as a proxy for increased attention (McGinley et al., 2015), resulted in increased information (as shown previously by (Dadarlat & Stryker, 2017)) and lower thresholds (Fig. S3).”

Additionally, we have added the following to the Discussion:

“First, the mouse’s state of arousal, commonly assessed by their pupil dilation, has been found to fluctuate during similar experiments (McGinley et al., 2015), and such fluctuations could modulate information encoded in V1. Locomotion is linked to arousal (McGinley et al., 2015), and has previously been shown to impact information (Dadgarlat & Stryker, 2017). In our data, periods of increased locomotion also result in more information in the recorded populations and increase asymptotic information estimates, but do not significantly affect the estimated population sizes required to encode 95% of this asymptotic information (Fig. S3).”

4. The behavioral state of the animal matters, but these mice were not doing any task and nothing was mentioned about pupil size or anything else that might signal changes in motivation or alertness, etc. The authors do discuss this factor related to the primate literature (Marlene Cohen’s work), but never in the context of how it really affects their estimates of redundancy and extrapolation. It is not only that attention reduces correlated variability, which is what was discussed. The mouse could be changing from “attentive” to “non-attentive” from trial-to-trial.

This is a good point that was also raised by Reviewer 2 (first two minor comments), and, indeed, we cannot directly assess attention. However, behavioral state can be co-modulated with locomotion, which is commonly used as a proxy for arousal and attention (e.g., Mineault et al., 2016). And indeed, we found increased sensory information for higher running speeds, as described above. Our analysis was performed across trials, and did not attempt to analyse running speed changes within trials - for this we neither had enough trials, nor sufficient temporal resolution in our neural recordings. However, given that a split into low/high running speed didn’t qualitatively affect our results, we would neither expect them to change in a more fine-grained analysis.

To alert the reader to the eventual impact of changes of arousal on our information measure and results, we have added some sentences about this to the Discussion (see response to preceding comment).

5. Orientation-dependent adaptation and offset responses will lead to substantial trial-to-trial variability. When randomly interleaving trials, you introduce variability because the response will be in a different state for every trial purely based on the previous stimulus presented. This is especially concerning because trials and inter-trials were only 500 ms in duration. Offset responses are almost certainly going to run into the next trial. This is especially true with GCaMP. I have seen offset responses last over 2 seconds even with static stimuli let alone drifting gratings. The effects of adaptation last for several seconds. The orientation biases in mouse will likely cause this variability to be correlated across neurons as well.

This is a great point that we haven’t addressed in the previous version of the manuscript. Adaptation might indeed be an issue for high-contrast stimuli that are presented in rapid

succession. Note, however, that we have used low-contrast stimuli for which adaptation will be significantly weaker, and should be non-consequential for our analysis. Furthermore, we attempted to compensate for slow GCaMP responses by performing our analysis after temporally deconvolving these responses (see main text and Methods).

We performed additional analysis to ensure that our data did not suffer from significant adaptation effects. Specifically, we asked if a neural response model that assumes that responses are driven by both the current and immediately preceding stimulus is able to explain the data better than a model that assumes these responses to only depend on the current stimulus. The results of this analysis are shown in new Table S1, and reveal no significant impact of drift direction-dependent orientation. We highlighted this at the beginning of Results by adding the following sentence:

“We found no significant impact of the drift direction in the previous trial on neural responses in the current trial (Fig. S1b and Table S1).”

6. I am also concerned about using such low contrast stimuli for mice. Many times tuning curves measured at 10% just look like a bunch of noise compared to tuning curves measured in the same neurons with 50% contrast. If you combine a very weak stimulus drive with all the factors above, correlated variability is going to dominate the responses. What are the R-squared measurements for your tuning curve fits? Are tuning curves consistent between 10 and 25%? What does Fig. 3 look like with the 25% contrast data?

Thank you for pointing out this potential issue. Having previously been able to categorize neurons into untuned, orientation-tuned, and direction-tuned despite the use of low contrast stimuli made us confident that our data wasn't simply driven by noisy measurements. Nonetheless, we have now performed the additional analysis suggested by the reviewer, and provide its results in the new Figs. S1 and S2. For our 10% data, we found that our fitted tuning curves could explain on average ~75% of the variance in the raw mean responses to each drift direction (see Fig. S1c).

When comparing 10% contrast trials to 25% contrast trials, we found an overall information increase, as reported in Fig. 6. However, we are unsure which parts of Fig. 3 the reviewer is asking us to replicate for the 25% contrast data, as those details are already provided in Fig. 6. To check if tuning curves are consistent between 10% and 25% contrast trials, we asked if assuming that the two tuning curves are simply scaled and shifted versions of each other provides a better fit than fitting them separately. Our analysis, which is now discussed in the new Fig. S10, revealed that a joint fit indeed explains the data better for most neurons, suggesting that the tuning curves are consistent across contrast levels for at least those neurons. Furthermore, we did not observe any strong differences in pairwise noise correlations between the two contrast levels (Fig. S10c).

We made this clear by adding the following sentences to the Section “A finite-population information change impacts asymptotic information”:

“For most neurons, a contrast increase from 10 to 25% led to a change in baseline activity and re-scaling of their tuning curves, but no appreciable change in pairwise noise correlations (Fig. S10). As in correlated populations we cannot predict changes in information solely from changes in tunings, we again moved to measuring information by our generalized Fisher information measure.”

7. I would also like to see Fig. 3 replotted with all of the manipulations done in Fig. 5. This leads me to another overall general major concern. This relates to the whole premise of the paper. Is there ever anywhere in the brain that is looking at all of VI at the same time? I would think that population coding (whether it is pooling or whatever) would always be from the perspective of particular subgroups. So why not look at what shuffling does to the optimal group of neurons for doing just one task? I would think that would be more relevant to what the brain is likely to implement if the mouse were to actually perform the task. The ideal group of neurons would presumably develop more highly weighted connections to sensorimotor areas related to the task through training. For example, 0-degree tuned neurons would be linked to “lick left” and 45-degree tuned neurons would be linked to “lick right”. I would think that the neurons unrelated to the discrimination task are unlikely to be used for decoding, since they will not even respond to the stimulus. If these neurons have no information about the stimulus, all they are contributing is correlated variability. Specific subgroups of neurons seems more related to mouse behavioral tasks and would be consistent with Newsome-style micro-stimulation results as well. If there is something I am missing, this needs to be discussed in more detail.

Thanks a lot for the comments and thoughts. We realize that our motivation for choosing specific drift direction pairs might have not been sufficiently clear in the previous version of the manuscript. Even though we measured information by focusing on the discrimination of specific drift direction pairs, we did not want to suggest that downstream areas ought to “read out” information about such specific discriminations. Specific discrimination tasks are the frequent focus of experimental work for practical reasons, and after extensive training might lead to over-specialization of cortical circuits to dealing with these discriminations only. This stands in contrast to natural animal behavior that rarely - if ever - requires such specialization. In fact, in natural circumstances such specialization might degrade behavioral performance. For this reason, we didn’t restrict ourselves to a single drift direction pair, but a larger range of pairs that act as a proxy for information about drift direction in general. This was not sufficiently clear in the manuscript, and we have added more clarification to the updated version (see below).

If we could restrict our attention to specific pairs of drift directions, we could indeed recover a large fraction of the information from smaller subsets of the population, as Fig. 5a shows. In such circumstances, our population size estimates to capture 95% of asymptotic information would also be inflated. Due to presence of orientation columns in monkeys, which are absent in mice, the subpopulation in monkeys might be even smaller, and anatomically more constrained. However, as laid out above, we tried to stay away from such over-specialization, and therefore

asked how well subpopulations convey information about a larger range of drift direction pairs - leading to our findings of Fig. 5b.

Our previous analysis did not show how much noise correlations contribute to our finding in Fig. 5b. This is an interesting question, as neurons can be untuned, but nonetheless contribute to information through noise correlations (e.g., Leavitt et al., 2017). Thus, as suggested by the reviewer, we re-did our analysis in Fig. 5 with trial-shuffled data. This new analysis is now shown in Fig. S9, and revealed that, without noise correlations, smaller subpopulations are indeed able to capture an increased fraction of the total information in the recorded population across all different tested discriminations. We have added this new result to section “No neural subpopulation encodes a disproportionate amount of information across all stimulus drift directions”:

“Noise correlations contribute to the observed lack of difference, as this difference becomes significant for trial-shuffled data (Fig. S9).”

More generally, we have clarified that our aim was to quantify information about drift direction in general, rather than about specific drift direction pairs. We have added the following to Section “Noise correlations limit information”:

“Importantly, our aim was to measure information that population activity conveyed about drift direction in general, without prioritizing specific drift directions over others. Even though subselecting a limited set of drift directions is common in animal training, we here focused on discriminating drift directions in pairs only as a tool to get at information about drift direction in general, which should be more reflective of real-world demands.”

Furthermore, we now remind the reader of this aim in Section “No neural subpopulation encodes a disproportionate amount of information across all stimulus drift directions” by adding

“However, natural behavior usually requires information about a wide range of different drift directions rather than the ability to discriminate a specific drift direction pair. To identify how much information the discovered subpopulation contains about other drift directions, we asked how well its population activity supports discriminating another, close-by drift direction pair (Fig. 5a; left vs. right). We found that the same subset of neurons was only able to recover about 55% of the information about this new discrimination.”

Minor Concerns:

1. Using drift direction instead of orientation can be a little confusing. Generally, orientation tuning and direction selectivity are the two descriptions of gratings, where the latter is always referring to 180-

degree differences in direction. With random dot noise, direction of motion is not linked to the orientation of the stimulus.

We agree that drift direction might be initially confusing to some readers. However, given the frequency with which the term appears in the manuscript, we didn't find an equally concise and unambiguous alternative. We hope that the reviewer understands that we chose not to separate drift direction into orientation and motion direction as it, in our opinion, (i) would have made the text more confusing and (ii) is not always unambiguous (e.g., if motion direction refers to left/right motion, we would not be able to distinguish up/down drift direction).

2. Fig 5a, left (0-degree grating has wrong orientation)

Thank you, fixed.

3. There is a paragraph labeled discussion in the methods that seems to be out of place?

This paragraph refers to additional analyses whose results are provided in the manuscript's Discussion section. We agree that labeling it simply 'Discussion' might be confusing and so have changed the label to 'Additional analyses in Discussion'

REVIEWER COMMENTS

Reviewer #1 (Remarks to the Author):

The authors have addressed the points raised in the previous revision. While extrapolating from finite amounts of data is a fraught enterprise, we currently lack technologies to densely record from the majority of neurons in a given area. Therefore, I believe this manuscript provides an important step toward understanding how information is represented in populations of neurons. The limitations of this approach are adequately addressed by the authors in the discussion.

Typo:

Line 236: "estimates" should be "estimated"

Reviewer #2 (Remarks to the Author):

The authors have done a commendable job of responding to my critiques. I have no further comments.

Congrats; this is a really nice study!

Joel Zylberberg

Reviewer #3 (Remarks to the Author):

While the authors made several additions to address many of my concerns (e.g., running and adaptation), some of my questions were not adequately addressed and in particular, I do not think questions about eye movements and trial-to-trial variance was sufficiently answered. Additionally, I share concerns raised by reviewers 1 and 2 about using GCaMP calcium imaging as a measurement of neural activity, variability, and information.

Mouse eye movements are substantially larger than the authors suggest. Changes in gaze are on average 10-12 degrees (Sakatani & Isa 2007; Samonds et al. 2018; Poort et al. 2020). However, it is not just about whether or not a change in gaze happens during a trial, but whether the eye position is different for different trials. Even if the gaze changes on average once every few seconds, the eyes can be positioned at different locations for different trials over a 40-degree range. This range is similar across most mammals. Gaze can change by as much as 30+ degrees and sometimes does change often (several times per second), especially during bursts of running. The argument that vision is so terrible in mice that monitoring eye movements is unnecessary seems like a poor argument to make in a submission about measuring performance of mouse vision. Yes, due to their relatively poor vision, receptive fields are large at an average diameter of 15 degrees, but can vary from being as small as 5 degrees to as large as 40+ degrees with surround modulation extending out to 90 degrees. Eye movements might also depend on the direction of motion of the stimulus as well, since shifts in gaze potentially move towards objects of interest (Michaël et al. 2020, Biorxiv). Response variability as a result of eye movements might also depend on direction of motion because eye movements are horizontally biased. Lastly, retinotopy can have direction preference biases that would cause correlated variability issues for direction coding as a result of eye movements. Overall, eye movements might move some receptive fields outside or partially outside of the stimulus for some trials or vary surround modulation. Since V1 neurons generally have simple receptive fields, even a shift involving part of the receptive field can have consequences. Averaging a response over an entire sinusoid cycle certainly helps to potentially remove variability, but not necessarily if the cell is biased for dark or light polarity and the slow dynamics of calcium vary the average response depending on whether the cycle hits the receptive field early or late during the short stimulus trial. I think measuring

visual responses in mice without monitoring eye movements is reasonable in cases where large stimuli are used with a very local representation of receptive fields and measuring simple response properties represented uniformly across space, such as orientation/direction tuning. In these cases, the variance does not substantially affect the tuning preference or shape and the error bars will just be larger than what actually represents neural trial-to-trial variability. When measuring visual response properties such as precise spatial receptive field properties, however, or in this case, properties that explicitly depend on trial-to-trial variability, this makes me much more skeptical about the results without being able to directly rule out the influence of eye movement and position.

I do think the running results are helpful and an interesting complement to the primary result in Fig. 3. Those results should be included in the main paper and demonstrate potentially how variability directly varies information estimates. I still have substantial concerns with these results though. Primarily, running increases response gain and eye movements together. It would be much better if you could separate out changes in eye movements and gain. You could potentially compare running and no running data using cells with matched responses. Therefore, you would at least remove gain as a factor. Additionally, if running is a proxy for attention, running might also reduce correlation like what is observed with attention in non-human primates. Here you could use only populations with matched correlation statistics, if correlation is indeed reduced during running. It would be nice to separate out these three factors dependent on running because there clearly is a difference in the influence of correlation between running and not running. Why is the Fisher Information higher and why does shuffling not reduce Fisher Information as substantially during running as compared to not running? More or less eye movements? Changes in gain? Changes in correlated variability? All three?

I have several concerns about using GCaMP to measure how correlated variability influences direction discrimination using neural populations. I asked to see raw data to understand the timing and trial-to-trial variability of the underlying data used in the information analysis. I still have not seen a single calcium trace or a measurement of variance. I want to know how much the variance is compared to the mean. The examples in Fig. S1 are concerning. If this is impossible to measure, how are we supposed to trust the quality of the data and the subsequent analysis? Is the GCaMP response variability comparable to electrophysiological measurements? Spike count correlation is not a fair replacement for direct measures of variance. It would be nice to see how the GCaMP timing is related to stimulus onset and offset over a series of presentations as well.

I did not find the answer to reviewer 1 with regard to neuropil to be sufficient. The neuropil is not uniform across the surface. The neuropil represents the activity of local neurons (not necessarily shared or already included in the population) and even though there are not orientation columns, direction selectivity can be biased in local regions (Ringach et al., 2016, Scholl et al., 2017; Fahey et al. 2019, Biorxiv). This could cause cells with weaker GCaMP responses to be artificially tuned for direction. This might also lead to artificial pooling that reduces the variance of single cell responses. The local neuropil with GCaMP6s viral injections can produce delta f/f responses as high as 0.05-0.1 which is well within the responses in Fig. 2. Why not directly address the reviewer's question by removing weaker responses that are potentially contaminated or adding local non-cell (but cell sized) neuropil regions to the analysis to see if it changes results?

I also did not find the answer to reviewer 2's question about the deconvolution of GCaMP responses into spike rates to directly address the question. The authors focus on the ability of GCaMP6s to detect single spikes, but reviewer 2 was concerned about resolving multiple spike activity into individual spikes. I am also highly concerned about how the relationship between spikes and calcium activity affects measurements of variance and information. When spike rates are low (1-2 spikes/s), as they are in many mouse V1 cells, the transformation is straightforward. But some cells do fire 20 spikes/s and running behavior might be increasing responses by double or more. Deconvolution will likely be able to separate out 1 versus 2 spikes and adequately capture trial-to-trial variance for low spike rates, but is this true for high spike rates? Can

deconvolution separate out if 7, 10, or 13 spikes occur within a single trial? Trial-to-trial variance is typically much easier to see with low spiking responses, but as the spike rates increase, the temporal profiles of GCaMP responses start to all look the same. Again, is the variance of deconvolved GCaMP responses consistent with known electrophysiology?

Response to reviewers

We would like to thank the reviewers for the careful examination of the changes in our manuscript and for their encouraging feedback.

Reviewer #1 (Remarks to the Author):

The authors have addressed the points raised in the previous revision. While extrapolating from finite amounts of data is a fraught enterprise, we currently lack technologies to densely record from the majority of neurons in a given area. Therefore, I believe this manuscript provides an important step toward understanding how information is represented in populations of neurons. The limitations of this approach are adequately addressed by the authors in the discussion.

We would like to thank the reviewer for the positive feedback and assessment of our manuscript.

Typo:

Line 236: “estimates” should be “estimated”

We could not identify a typo on line 236, but have instead fixed one on line 215 (in the revised version of the manuscript) that matches the above description. We hope that this is the typo that the reviewer had in mind.

Reviewer #2 (Remarks to the Author):

The authors have done a commendable job of responding to my critiques. I have no further comments.

Congrats; this is a really nice study!

Joel Zylberberg

Thank you, Joel, for your positive feedback and your appreciation of our work.

Reviewer #3 (Remarks to the Author):

While the authors made several additions to address many of my concerns (e.g., running and adaptation), some of my questions were not adequately addressed and in particular, I do not think questions about eye movements and trial-to-trial variance was sufficiently answered. Additionally, I share concerns raised by reviewers 1 and 2 about using GCaMP calcium imaging as a measurement of neural activity, variability, and information.

We appreciate the reviewer’s careful consideration of our revisions and previous response to reviewers. In the updated manuscript and below responses we hope to have addressed the remaining concerns.

Mouse eye movements are substantially larger than the authors suggest. Changes in gaze are on average 10-12 degrees (Sakatani & Isa 2007; Samonds et al. 2018; Poort et al. 2020). However, it is not just about

whether or not a change in gaze happens during a trial, but whether the eye position is different for different trials. Even if the gaze changes on average once every few seconds, the eyes can be positioned at different locations for different trials over a 40-degree range. This range is similar across most mammals. Gaze can change by as much as 30+ degrees and sometimes does change often (several times per second), especially during bursts of running. The argument that vision is so terrible in mice that monitoring eye movements is unnecessary seems like a poor argument to make in a submission about measuring performance of mouse vision. Yes, due to their relatively poor vision, receptive fields are large at an average diameter of 15 degrees, but can vary from being as small as 5 degrees to as large as 40+ degrees with surround modulation extending out to 90 degrees. Eye movements might also depend on the direction of motion of the stimulus as well, since shifts in gaze potentially move towards objects of interest (Michaël et al. 2020, Biorxiv). Response variability as a result of eye movements might also depend on direction of motion because eye movements are horizontally biased. Lastly, retinotopy can have direction preference biases that would cause correlated variability issues for direction coding as a result of eye movements. Overall, eye movements might move some receptive fields outside or partially outside of the stimulus for some trials or vary surround modulation. Since V1 neurons generally have simple receptive fields, even a shift involving part of the receptive field can have consequences. Averaging a response over an entire sinusoid cycle certainly helps to potentially remove variability, but not necessarily if the cell is biased for dark or light polarity and the slow dynamics of calcium vary the average response depending on whether the cycle hits the receptive field early or late during the short stimulus trial. I think measuring visual responses in mice without monitoring eye movements is reasonable in cases where large stimuli are used with a very local representation of receptive fields and measuring simple response properties represented uniformly across space, such as orientation/direction tuning. In these cases, the variance does not substantially affect the tuning preference or shape and the error bars will just be larger than what actually represents neural trial-to-trial variability. When measuring visual response properties such as precise spatial receptive field properties, however, or in this case, properties that explicitly depend on trial-to-trial variability, this makes me much more skeptical about the results without being able to directly rule out the influence of eye movement and position.

We would like to thank the reviewer for highlighting the potential impact of eye movements on our information measures. As mentioned in our previous reply to reviewers, we did not collect eye movement data during our experiments, and so cannot directly analyze how they might impact population responses and associated information measures. However, we would like to draw attention to the fact that eye movement statistics depend on details of the experiments for which they were collected. For the papers the reviewer mentioned, mice were either in complete darkness (Sakatani & Isa, 2007), or freely moving / during prey capture (Meyer, O'Keefe & Poort, 2020; Michaël et al., 2020), both of which are situations that are very different from our experiments in which mice were head-fixed and observed full-field drifting gratings. Samonds et al. (2018) used head-fixed mice, but presented natural stimuli. More relevant papers might be Andermann, Kerlin & Reid (2010) and Ayaz et al. (2013), in which mice passively viewed drifting gratings and featured much smaller pupil displacements of ~5deg, which also occurred infrequently. Similarly, in Keller, Bonhoeffer & Hubener (2012), in head-fixed mice viewing drifting full-field gratings controlled by their running, eye movements were also ~5deg and were rare (~0.12-0.3 saccades/sec). In Poort et al. (2015), head-fixed mice trained to discriminate between two grating orientations had small eye movements (<10deg, mean of 0deg) that occurred infrequently, with saccade rates of 0.05-0.1 Hz. Speed et al. (2020) also found exceedingly small pupil displacements (<2deg) in mice viewing gratings in a spatial attention task. Overall, these papers suggest that eye movement is significantly smaller if mice are head-fixed and/or presented with drifting gratings, as is the case in our experiments, with eye movement statistics in line with those provided in our previous response to reviewers.

In this context we would also like to re-emphasize that it isn't variability and the presence of noise correlations per se that limit information. Instead, it is variability and co-variability of a very specific structure, namely that of differential correlations (Moreno-Bote et al., 2014) that induce noise in the same direction as the signal of interest. In our experiment the signal of interest is the direction of a drifting grating, which is a circular variable. We would expect changes in gaze to instead introduce variability in linear directions. Therefore, even if changes of gaze introduce additional neural variability, they might not have a large impact on our estimates of information of a circular variable (they might impact tasks in which the discrimination is linear, e.g., the random-dot motion task). Showing this in simulations is unfortunately beyond the model of Kanitscheider et al. (2015) that we have used, as that model assumes the center of stimulus rotation to be aligned with the neural receptive fields. It is already computationally expensive to simulate the model for a single such gaze location, and extending it to multiple gaze locations would be computationally prohibitive. However, we now illustrate the difference between total variability and variability that limits information in **Fig. S3**, that we discuss in more detail further below.

Overall, we are confident that eye movements in our experiments do not qualitatively change the conclusions of our work. Previous literature with experimental setups comparable to ours suggests eye movements to be relatively small and infrequent. Furthermore, for the theoretical reasons discussed above and in the Discussion, even if present, we expect their impact on our information estimates to be fairly minimal. We hope that the reviewer agrees with this assessment.

I do think the running results are helpful and an interesting complement to the primary result in Fig. 3. Those results should be included in the main paper and demonstrate potentially how variability directly varies information estimates. I still have substantial concerns with these results though. Primarily, running increases response gain and eye movements together. It would be much better if you could separate out changes in eye movements and gain. You could potentially compare running and no running data using cells with matched responses. Therefore, you would at least remove gain as a factor. Additionally, if running is a proxy for attention, running might also reduce correlation like what is observed with attention in non-human primates. Here you could use only populations with matched correlation statistics, if correlation is indeed reduced during running. It would be nice to separate out these three factors dependent on running because there clearly is a difference in the influence of correlation between running and not running. Why is the Fisher Information higher and why does shuffling not reduce Fisher Information as substantially during running as compared to not running? More or less eye movements? Changes in gain? Changes in correlated variability? All three?

We thank the reviewer for their interest in running speed analysis that we added to the previous revision of the manuscript. As this analysis is based on a running speed median split, and not on any neural measures, it does not directly demonstrate how variability might impact our information estimate. This makes it less central to the main point of the manuscript. Furthermore, an increase of information with running speed has been reported before (e.g., Dadarlat & Stryker, 2017), which makes our analysis less novel. Therefore, we would prefer to keep it in the supplement rather than including it as one of the main figures. We hope the reviewer agrees.

As already mentioned, we do not have eye movement data available, and so cannot analyze the contribution of eye movements to the information increase due to increased running speed. However, we were able to analyze which neural population response features cause this information increase. To do so, we considered the impact of a running speed-related change on

neural tuning (i.e., a change in mean population responses) and noise covariance. We found that it wasn't one of these factors in isolation, but both in combination that resulted in the information increase with increased running. For a further understanding of the impact of a changing noise covariance, we split its contribution into a change in per-neuron response variance, as well as a change in noise correlations (rather than covariance) structure. We again found that both contribute positively to the observed information boost. Therefore, the information boost couldn't be ascribed to a change in mean population response, in per-neuron noise variance, or in pairwise noise correlations alone, but was a result of a change of all of these factors. This replicates previous findings of Dadarlat & Stryker (2017; Fig. 5) that used electrophysiological recordings to demonstrate that a change in both tuning and noise covariance contributes to the information boost for higher running speeds. Being able to replicate these electrophysiological results with calcium imaging data also confirms that calcium imaging yields useful data to assess the information encoded in population activity.

To describe these results, we have added **Fig. S5** to the manuscript. Furthermore, we added the following sentence to the Section "Noise correlations limit information":

"In line with previous findings (Dadarlat & Stryker, 2017), this information boost was caused by a combination of a change in population tuning, per-neuron noise variability, and pairwise noise correlations, rather than by either of these factors in isolation (Fig. S5)".

Due to data sparsity, we were unable to subsample trials, as suggested by the reviewer. Instead, we exploited the fact that our expression for information naturally decomposes into the investigated factors, which allowed us to assess their impact when varying them in isolation, as we did in this figure. This yielded the results outlined above.

I have several concerns about using GCaMP to measure how correlated variability influences direction discrimination using neural populations. I asked to see raw data to understand the timing and trial-to-trial variability of the underlying data used in the information analysis. I still have not seen a single calcium trace or a measurement of variance. I want to know how much the variance is compared to the mean. The examples in Fig. S1 are concerning. If this is impossible to measure, how are we supposed to trust the quality of the data and the subsequent analysis? Is the GCaMP response variability comparable to electrophysiological measurements? Spike count correlation is not a fair replacement for direct measures of variance. It would be nice to see how the GCaMP timing is related to stimulus onset and offset over a series of presentations as well.

We are sorry that we have misinterpreted the reviewer's previous request (we added **Fig. S1** in the previous response, assuming that the temporal averages were enough). We have now added an additional **Figure S3** that shows raw dF/F traces for some example neurons (both 10% and 25% contrast examples; note that to measure information, we didn't use the shown raw traces, but temporally deconvolved versions thereof). In addition to showing the raw traces, we furthermore show them projected onto the optimal decoder. This projection re-emphasizes our main point that it isn't variability (and co-variability) per se that limits information, but only a specific kind of variability pattern called *differential correlations*. As the new **Fig. S3** shows, the across-trial variability of dF/F traces almost completely disappears once these traces are projected onto the optimal decoder (see also Montijn et al., 2016, which makes a similar point). This implies that most of the raw traces' variability does *not* limit information. We now highlight this additional figure in the first Results paragraph:

“See **Figs. S1-S3** for more examples of neural responses, tuning curves, pairwise noise correlations, and raw calcium traces.”

Furthermore, we point the reader to it in the Discussion:

“In general, only those factors that modulate information-limiting correlations, which are a small component of the overall noise correlation matrix, impact our information estimates (illustrated in **Fig. S3**).”

We furthermore show some longer example traces in **Fig. R1** below, to illustrate the GCaMP response time-course over multiple stimulus presentations. Note that we temporally deconvolved these traces before using them to estimate information.

Figure R1. Longer dF/F traces for four example neurons.

The colored bars in the background show the stimulus presentation period (colors/arrows above bars = drift directions). The contrast of the arrows above these bars indicate the respective stimulus' contrast. The top two neurons are from single-contrast sessions, the bottom two neurons from dual-contrast sessions. All show the GCaMP response over a 50s period.

Some measures of variability (e.g., Fano factor) cannot be directly computed from calcium imaging data because fluorescence responses are scaled by some unknown, arbitrary factor relative to spiking activity. For example, let us assume that some neuron features spike count r (a number that fluctuates across trials), and that the measured pre-trial GCaMP responses x are some linear rescaling of r , that is $x = \alpha r$, with an unknown scaling factor α that might vary across neurons (see below for the impact of *nonlinear* rescaling). This results in the variance of x to be given by $var(x) = \alpha^2 var(r)$, from which we cannot unambiguously recover the spike count variance $var(r)$. The same principle applies to the Fano factor. Computed from GCaMP

responses, we find $\text{var}(x)/\langle x \rangle = \alpha^2 \text{var}(r)/(\alpha \langle r \rangle) = \alpha \text{var}(r)/\langle r \rangle$. This shows that the Fano factor computed from GCaMP responses is the one computed from spike counts scaled by an unknown factor, making it uninformative. Therefore, we refrained from adding these measures to the manuscript.

However, other measures of variability are not impacted by this arbitrary scaling factor. The coefficient of variation (CV), for example, is the same when computed from x as when computed from r , that is, $SD(x)/\langle x \rangle = \alpha SD(r) / (\alpha \langle r \rangle) = SD(r)/\langle r \rangle$. In our data, we found this CV to be roughly one on average (see **Fig. R2** below). This CV is in line with previous mouse V1 data. Bennett, Arroyo & Hestrin (2013), for example, found in whole-cell patch clamp recordings a CV of between ~1 (moving) to 2 (stationary) in response to drifting sinusoidal gratings. de Vries et al. (2020) found a higher CV of ~2.5 from two-photon calcium imaging data in response to drifting gratings. Therefore, the CV of our recordings is on the lower side (i.e., less variable) when compared to previous recordings, including most electrophysiology recordings.

Most importantly, arbitrary re-scaling does not impact our information measures. This measure is invariant to invertible linear transformations of the data, such that it is unaffected by such re-scaling, even if the scaling factor varies across neurons.

Figure R2. Coefficients of variation for direction- and orientation-tuned neurons.

We computed the coefficient of variation (CV; SD / mean) for direction- and orientation-tuned neurons for the session shown in main text **Fig. 3**. The CV was computed for each neuron from deconvolved calcium traces, summed over the stimulus presentation period, across all trials for the drift direction that evoked the on average largest neural response for that neuron. Lines show the cumulative fraction for different CVs across neurons, and vertical bars show the means. We found comparable distributions and means across all sessions (not shown).

I did not find the answer to reviewer 1 with regard to neuropil to be sufficient. The neuropil is not uniform across the surface. The neuropil represents the activity of local neurons (not necessarily shared or already included in the population) and even though there are not orientation columns, direction selectivity can be biased in local regions (Ringach et al., 2016, Scholl et al., 2017; Fahey et al. 2019, Biorxiv). This could cause cells with weaker GCaMP responses to be artificially tuned for direction. This might also lead to artificial pooling that reduces the variance of single cell responses. The local neuropil with GCaMP6s viral injections can produce delta f/f responses as high as 0.05-0.1 which is well within the responses in Fig. 2. Why not directly address the reviewer's question by removing weaker responses that are potentially contaminated or adding local non-cell (but cell sized) neuropil regions to the analysis to see if it changes results?

As suggested by the reviewer, we have now performed additional analyses that investigate whether weakly responding neurons perturb our results. The results of these analyses are shown in **Figure R3** (see below) and confirm our previous suggestions that neuropil contamination is unlikely to qualitatively change our findings. In particular, we found that weakly responding neurons do not contribute more or less information than other neurons in the population. Furthermore, we found that removing them did not change our conclusion that no neural subpopulation encodes a disproportionate amount of information across all stimulus drift directions (as previously asked by reviewer 1).

Figure R3: removing low-response neurons does not qualitatively change our results.

(a) As the histogram of measured response intensity (F) across all sessions shows, we did not observe a clearly identifiable separate set of lower intensities. We observed the same continuity of intensities within each session (not shown).

(b) To see if removing low-response neurons results in a significant change of information, we compared the population information when removing the 10, 20, 30 or 40 (colors) lowest-responding neurons to the information

when removing the same number of randomly selected neurons for each 45deg discrimination and dataset. We determined statistical significance from bootstrap information samples by removing different sets of randomly selected neurons. The plot shows the cumulative density function of these p-values across all such discriminations and datasets. We found that the fraction of discriminations/datasets for which this p-value is below 0.05 does not exceed the one expected by chance (binomial test). Therefore, we did not find a significant change of information. (c) To test if removing low-response neurons impacts the analysis shown in **Fig. 5b** in the main text, we replicated this analysis after removing the 10, 20, 30 or 40 lowest-responding neuropils (fill dots = significant difference for respective discrimination). The required population size to capture 90% of information for random neuron orders vs. optimized orders did not differ significantly for either number of removed neurons (two-sided t-test, $p > 0.147$). Therefore, the results of our analysis are not affected by removing low-response neurons.

I also did not find the answer to reviewer 2's question about the deconvolution of GCaMP responses into spike rates to directly address the question. The authors focus on the ability of GCaMP6s to detect single spikes, but reviewer 2 was concerned about resolving multiple spike activity into individual spikes. I am also highly concerned about how the relationship between spikes and calcium activity affects measurements of variance and information. When spike rates are low (1-2 spikes/s), as they are in many mouse V1 cells, the transformation is straightforward. But some cells do fire 20 spikes/s and running behavior might be increasing responses by double or more. Deconvolution will likely be able to separate out 1 versus 2 spikes and adequately capture trial-to-trial variance for low spike rates, but is this true for high spike rates? Can deconvolution separate out if 7, 10, or 13 spikes occur within a single trial? Trial-to-trial variance is typically much easier to see with low spiking responses, but as the spike rates increase, the temporal profiles of GCaMP responses start to all look the same. Again, is the variance of deconvolved GCaMP responses consistent with known electrophysiology?

The impact of saturating mapping between spike counts and GCaMP responses is indeed an interesting question that would affect, among a wide range of other work, the vast literature that applies linear decoders directly to GCaMP responses. Due to a lack of simultaneous electrophysiology and calcium imaging recordings, we cannot use our data to make direct statements about the presence and degree of such saturation. However, we can address the impact of such a saturating mapping in principle, through simulations and theory.

For the purpose of our work, it is most important that such a saturating nonlinearity does not qualitatively change how information scales with population size. Fortunately, such nonlinearities do not make uncorrelated neurons correlated, and do not remove noise correlations. Most importantly, they don't introduce noise in the signal direction (i.e., differential correlations) if it wasn't there in the first place, and don't remove it if it is present. Therefore spike count information that saturates due to noise correlations also saturates if spike counts are non-linearly perturbed. We have added a new **Fig. S14** that shows this for saturating nonlinearities of multiple strengths. Furthermore, we have added the following sentence to the Discussion:

“Furthermore, saturation of GCaMP responses might have caused a non-linear mapping between spike counts and measured GCaMP responses, which would quantitatively lower the measured information, but not qualitatively impact how information scales with population size (Fig. S14).”

REVIEWERS' COMMENTS

Reviewer #1 (Remarks to the Author):

As before, I think the authors have done a commendable job and the study constitutes an important advance. Using simulations to examine the potential impact of eye movements on estimates of asymptotic information and the number of neurons necessary to capture 95% of total information would strengthen the manuscript. However, my intuitions agree with the author's judgment that eye movements are unlikely to affect their conclusions.

I think the authors understate the typical magnitude of eye movements made by head-fixed mice, which are often in the range of +/- 10 degrees (e.g. Meyer, et al., *Current Biology*, 2020), comparable to the size of many mouse V1 receptive fields. Therefore, one would expect that eye movements might affect the fraction of neurons that respond to the visual stimulus. Since the grating stimulus was windowed in a 20 degrees radius circle in most experiments, some of the RFs may end up outside of the area covered by the grating. This would clearly decrease c , the amount of information contributed per neuron. I would expect that it would inflate estimates of the number of neurons needed to capture 95% of total information but would not affect estimates of asymptotic information as $N \rightarrow \infty$. If this is the case, it would not affect the manuscript's conclusions, including the author's claim that information saturates with numbers of neurons much smaller than those available in V1. Testing these intuitions by comparing simulations that include one vs multiple (even just two) gaze directions would strengthen the manuscript. If this is computationally prohibitive for the more realistic V1 model, would it be possible by e.g. drawing responses from the Gaussian population activity model and then setting the response of a subset of neurons to 0 on a fraction of trials to mimic the situation where their RFs fall outside of the grating area due to eye movement?

The discussion regarding the impact of running on information content of V1 population responses is an interesting one. In fact, there is now ample evidence that running and other movements affect neuronal responses in more complex ways than straightforward gain modulation and explain a substantial fraction of trial-to-trial variability in neuronal responses (e.g. Stringer, Pachitariu, et al., *Science*, 2019). How these movement-related modulations affect the information about the visual scene present in V1 responses remains an open question. However, to make judgments about the content of the visual scene, the rest of brain faces the challenge of decoding V1 activity in the presence of these modulations. Therefore, while the degree to which running and other movement-related activity contribute to information limiting correlations described in the manuscript is an interesting question, I think the conclusion that correlations limit information in V1 stands on its own.

The new Figure S3 is a great addition to the manuscript. The raw fluorescence traces show the degree of variability that I would expect from awake recordings in mouse V1. The projection of single trial responses onto the optimal decoder is very informative. Clearly, GCaMP fluorescence is not a perfect proxy for firing rate. While this may decrease the estimates of asymptotic information, it cannot explain the presence of information-limiting correlations described in the manuscript. Regarding review #3's further points on neuropil contamination, the authors did use a fairly rigorous procedure to control for neuropil fluorescence, although I am somewhat concerned that the CNMF model included only 3 background components for a fairly large field of view. In addition to making some unresponsive neurons appear responsive, one further way in which residual neuropil fluorescence could impact the presented analyses is by allowing information from neurons that were not imaged to "leak out" to the imaged cells. This could lead to an underestimate of the number of neurons needed to capture 95% of total information. Addressing this possibility would strengthen the discussion of the limitations of the imaging approach.

Reviewer #2 (Remarks to the Author):

The authors have once again done quite a thorough response to the reviewer's concerns. This has improved the paper, although I feel that the revisions are now in the domain of diminishing returns.

I remain supportive of this paper being published.

Joel Zylberberg

Reviewer #3 (Remarks to the Author):

I find the additional analysis to be sufficient and do not need to see any new revisions. I still have concerns, but they do not rise to the level to prevent publication. My criticisms arose primarily because I view this as an extrapolation of a model fit to a deconvolution of a nonlinear transformation of a highly irregular and dynamic signal. Even though many of these steps have become more common, it does not mean that they are not potentially problematic. Therefore, my main concern was that the authors make it clear about how all of those steps introduce some uncertainty, which is expressed in the discussion sufficiently. Doing that and demonstrating that the analysis involves carefully collected reliable data will make experimentalists much more comfortable in taking the conclusions seriously. I recommend including the CV analysis and discussion about the underlying data and experimental reliability in the methods.

Line 96 delete "a"

Line 97 delete "such"

Line 212 Could delete the parenthetical "(except, trivially, for single neurons)"

Line 228 Should briefly mention how your discrimination performance relates to known behavioral data (e.g., Andermann et al. 2010 or various related studies also mentioned briefly in the discussion).

Line 232-233 reword to "the estimated threshold likely underestimates discrimination capabilities"
Line 654-655 maybe add Nienborg & Cumming (2006).

Discussion: could add Andermann et al. 2010 to behavioral threshold comparisons. Besides task differences, mice might not integrate with as many cells as you predict and therefore performance might fall below your predictions. Or mice may use a completely different neural integration strategy. Or mice may not be motivated enough in behavioral measurements and their thresholds are much better.

REVIEWERS' COMMENTS

Reviewer #1 (Remarks to the Author)

As before, I think the authors have done a commendable job and the study constitutes an important advance. Using simulations to examine the potential impact of eye movements on estimates of asymptotic information and the number of neurons necessary to capture 95% of total information would strengthen the manuscript. However, my intuitions agree with the author's judgment that eye movements are unlikely to affect their conclusions.

We would like to thank the reviewer for insightful and constructive comments. In the updated manuscript and below responses we hope to have addressed the remaining concerns.

I think the authors understate the typical magnitude of eye movements made by head-fixed mice, which are often in the range of +/- 10 degrees (e.g. Meyer, et al., Current Biology, 2020), comparable to the size of many mouse V1 receptive fields.

Thank you for pointing us to Meyer et al. (2020). We would like to re-emphasize that eye movement magnitudes appear to be highly dependent on the exact experimental setup and the used stimuli. The magnitudes we have reported were from past work that was closest to our experimental setup and used similar stimuli (head-fixed mice, large drifting gratings). Past literature suggests that eye movement magnitudes increase in natural environments and in freely roaming animals (e.g., Meyer, O'Keefe & Poort, 2020; Michael et al., 2020). In Meyer et al. (2020), a subset of mice were head-fixed, but appeared to be otherwise freely running on a circular platform, but without any additional stimulation. It was unclear to us if experiments were performed in darkness or light. Either option leads to a visual environment that was significantly different from our experiments, such that we don't expect the reported eye movement magnitude to translate to our experiment.

Therefore, one would expect that eye movements might affect the fraction of neurons that respond to the visual stimulus. Since the grating stimulus was windowed in a 20 degrees radius circle in most experiments, some of the RFs may end up outside of the area covered by the grating. This would clearly decrease c , the amount of information contributed per neuron. I would expect that it would inflate estimates of the number of neurons needed to capture 95% of total information but would not affect estimates of asymptotic information as $N \rightarrow \infty$. If this is the case, it would not affect the manuscript's conclusions, including the author's claim that information saturates with numbers of neurons much smaller than those available in V1. Testing these intuitions by comparing simulations that include one vs multiple (even just two) gaze directions would strengthen the manuscript. If this is computationally prohibitive for the more realistic V1 model, would it be possible by e.g. drawing responses from the Gaussian population activity model and then setting the response of a subset of neurons to 0 on a fraction of trials to mimic the situation where their RFs fall outside of the grating area due to eye movement?

We thank the reviewer for the suggestions, and followed it by simulated data in which the activity of some neurons was set to zero in a fraction of trials. We found that, as the reviewer suspected, the simulated impact of such eye movements resulted in an overestimation of N_{95} , but only had a minor effect on the estimated I_{∞} . We furthermore confirmed these effects in a new theoretical analysis. The results of the simulations are shown in the new Supplementary Figure S14. The associated theoretical analysis is provided in the new Supplementary Information Sec. 2.4. We furthermore now point the reader to this information in the Discussion:

“This was confirmed in simulations and theoretical analysis of a simple eye movement model, which revealed that the assumed eye movements might result in over-estimating N_{95} , but only in a minor underestimation of I_{∞} (Fig. S14).”

The discussion regarding the impact of running on information content of V1 population responses is an interesting one. In fact, there is now ample evidence that running and other movements affect neuronal responses in more complex ways than straightforward gain modulation and explain a substantial fraction of trial-to-trial variability in neuronal responses (e.g. Stringer, Pachitariu, et al., Science, 2019). How these movement-related modulations affect the information about the visual scene present in V1 responses remains an open question. However, to make judgments about the content of the visual scene, the rest of brain faces the challenge of decoding V1 activity in the presence of these modulations. Therefore, while the degree to which running and other movement-related activity contribute to information limiting correlations described in the manuscript is an interesting question, I think the conclusion that correlations limit information in V1 stands on its own.

We appreciate the reviewer's comment and indeed agree with the reviewer's judgment about the impact of running speed on the information. It would be interesting to see if movement-related activity changes in V1 supports this movement, but this is clearly beyond the scope of our study.

The new Figure S3 is a great addition to the manuscript. The raw fluorescence traces show the degree of variability that I would expect from awake recordings in mouse V1. The projection of single trial responses onto the optimal decoder is very informative. Clearly, GCaMP fluorescence is not a perfect proxy for firing rate. While this may decrease the estimates of asymptotic information, it cannot explain the presence of information-limiting correlations described in the manuscript. Regarding review #3's further points on neuropil contamination, the authors did use a fairly rigorous procedure to control for neuropil fluorescence, although I am somewhat concerned that the CNMF model included only 3 background components for a fairly large field of view. In addition to making some unresponsive neurons appear responsive, one further way in which residual neuropil fluorescence could impact the presented analyses is by allowing information from neurons that were not imaged to “leak out” to the imaged cells. This could lead to an underestimate of the number of neurons needed to capture 95% of total information. Addressing this possibility would strengthen the discussion of the limitations of the imaging approach.

To model spatially and temporally neuropil fluorescence, we used 3 background components in our CNMF algorithm instead of 1 (the default number) to better capture any background contamination. Using 1-3 background components is standard for CNMF. For example, documentation for the Python implementation of CalmAn (https://caiman.readthedocs.io/en/master/Getting_Started.html) states that 3 background components can “fit for more complex noise” and that 1 is “usually too low,” and also notes that if the number of background components is too low then extracted traces are noisy, whereas if the number is too high then neuronal signals are absorbed into the background and transients are reduced. We thus believe that our choice of 3 background components to be careful about neuropil contamination is justified.

In addition, in our previous response to reviewers, we showed with additional analyses that weakly responding neuropil do not contribute more or less information than other neurons in the population (see previous **Figure R1**). Removing weakly-responsive neuropil did not change our results that no neural subpopulation encodes a disproportionate amount of information across stimulus drift directions. However, we cannot completely rule out leakage of information, and therefore have followed the reviewer’s suggestion and added an additional clarification to the Discussion:

"Moreover, residual neuropil fluorescence could cause the non-recorded neuron’s signal to “leak out” to recorded neurons, which might result in an underestimation of N₉₅."

Reviewer #2 (Remarks to the Author)

The authors have once again done quite a thorough response to the reviewer's concerns. This has improved the paper, although I feel that the revisions are now in the domain of diminishing returns.

I remain supportive of this paper being published.

Joel Zylberberg

We would like to again thank the reviewer for his appreciation of our work.

Reviewer #3 (Remarks to the Author)

I find the additional analysis to be sufficient and do not need to see any new revisions. I still have concerns, but they do not rise to the level to prevent publication. My criticisms arose primarily because I view this as an extrapolation of a model fit to a deconvolution of a nonlinear transformation of a highly irregular and dynamic signal. Even though many of these steps have become more common, it does not mean that they are not potentially problematic. Therefore,

my main concern was that the authors make it clear about how all of those steps introduce some uncertainty, which is expressed in the discussion sufficiently. Doing that and demonstrating that the analysis involves carefully collected reliable data will make experimentalists much more comfortable in taking the conclusions seriously. I recommend including the CV analysis and discussion about the underlying data and experimental reliability in the methods.

We would like to thank the reviewer for the positive feedback and assessment of our manuscript. To address the reviewer's comment we have updated the Method sections to include the coefficient of variation (CV) analysis and discussion about the underlying data and experimental reliability. The updated text read as follows:

"To assess neural variability in our recordings, we computed the coefficient of variation (CV; i.e., relative standard deviation) for orientation- and direction-tuned neurons. We found this CV to be roughly one on average, which compares favorably to previously reported mouse V1 data. Bennett, Arroyo & Hestrin (2013), for example, found in whole-cell patch clamp recordings a CV of between ~1 (moving) to 2 (stationary) in response to drifting sinusoidal gratings. de Vries et al. (2020) found a higher CV of ~2.5 from two-photon calcium imaging data in response to drifting gratings. As fluorescence responses are scaled by some unknown, arbitrary factor relative to spiking activity, we could not compute the neurons' Fano factors. This scaling did not impact our linear Fisher information estimates, as these estimates are invariant to (invertible) linear transformations of neural activity."

Line 96 delete "a"

Line 97 delete "such"

Line 212 Could delete the parenthetical "(except, trivially, for single neurons)"

Fixed, thank you.

Line 228 Should briefly mention how your discrimination performance relates to known behavioral data (e.g., Andermann et al. 2010 or various related studies also mentioned briefly in the discussion).

We would like to thank the reviewer for the suggestion. We have updated our manuscript to include the explicit discrimination performance numbers reported in the literature.

In the revised version, we have added the following sentences to the Results:

"Previously reported discrimination threshold of mice, as measured from behavioral performance, ranged from 6.6° (Glickfeld, Histed, & Maunsell, 2013) over 10-20° (Andermann, 2010), to 30-40° (Abdolrahmani, Lyamzin, Aoki, & Benucci, 2019). These numbers provide an orders-of-magnitude comparison, but cannot be directly compared to our estimate, as neither study exactly matched the stimuli we used. Moreover, [...]"

Line 232-233 reword to “the estimated threshold likely underestimates discrimination capabilities”

Updated the text, thank you.

Line 654-655 maybe add Nienborg & Cumming (2006).

Added the aforementioned reference to the manuscript, thank you.

Discussion: could add Andermann et al. 2010 to behavioral threshold comparisons. Besides task differences, mice might not integrate with as many cells as you predict and therefore performance might fall below your predictions. Or mice may use a completely different neural integration strategy. Or mice may not be motivated enough in behavioral measurements and their thresholds are much better.

We would like to thank the reviewer for the constructive comments throughout the review process. As mentioned above, we now report explicit discrimination performances measured in past works, including Andermann et al. (2010), in Results. Additionally, we have added the following to the Discussion:

"We furthermore can't exclude the possibility that mice used a different read-out than the linear one we assumed, or lacked motivation to perform the task to their full potential, further impacting their behavioral performance."